# Camel milk is a neglected source of brucellosis among rural Arab communities

Peter Holloway [1] ✉, Matthew Gibson[1], Tanja Holloway[2], Iain Pickett[1], Brittany Crook[1], Jacqueline M. Cardwell [1], Stephen Nash[3], Imadidden Musallam[1], Bilal Al-Omari[4], Ahmad Al-Majali[4], Wail Hayajneh[5], Ehab Abu-Basha[4], Punam Mangtani[2,6] & Javier Guitian[1,6]

The World Health Organization describes brucellosis as one of the world's leading zoonotic diseases, with the Middle East a global hotspot. *Brucella melitensis* is endemic among livestock populations in the region, with zoonotic transmission occurring via consumption of raw milk, amongst other routes. Control is largely via vaccination of small ruminant and cattle populations. Due to sociocultural and religious influences camel milk (*camelus dromedarius)* is widely consumed raw, while milk from other livestock species is largely boiled. To investigate the potential public health impact of *Brucella* in camels we conduct a cross-sectional study in southern Jordan including 227 herds and 202 livestock-owning households. Here we show daily consumption of raw camel milk is associated with *Brucella* seropositive status among the study population, OR$_{adj}$ 2.19 (95%CI 1.23–3.94) on multivariable analysis, highlighting the need for socioculturally appropriate control measures; targeted interventions among the camel reservoir being crucial for effective control.

With over two million human cases estimated to occur annually worldwide, the World Health Organization (WHO) has described brucellosis as one of the world's leading zoonotic diseases. Most cases occur within lower-and-middle-income countries (LMIC) with limited resources for prevention and control, prompting WHO to designate brucellosis a global Neglected Disease[1–3]. Human infection causes a variety of symptoms including recurrent fevers, headaches, arthralgia and chronic malaise, while reproductive symptoms also occur, including miscarriages, infertility and orchitis[4,5]. In addition, infection can also progress to cause osteomyelitis, endocarditis, meningoencephalitis, and in some cases death[4]. Human infections occur largely via consumption of infected raw milk and dairy products and through physical contact with infected livestock, via contaminated birthing products and reproductive discharges[6]. As such, brucellosis also represents an important production disease in livestock, resulting in

potentially severe economic losses through abortions, still-births, metritis, orchitis and infertility[6].

The Middle East represents an important global hotspot for brucellosis, with more than half a million human cases estimated to occur annually[7–9]. *B. melitensis* is the dominant causative agent (alongside, to a lesser extent, *B. abortus*), with reported livestock seroprevalences across the region being among the highest in the world[7,10,11]. The majority of human cases in the region occur among sub-populations working closely with livestock, with consumption of unpasteurized dairy products and occupational exposure to breeding ruminants being the most frequently identified risk factors for infection[8,12,13]. Within Jordan, brucellosis has been identified as one of the most important zoonotic diseases of public health impact, with government-led control strategies largely focussed on livestock vaccination schemes[14–18]. However, despite several decades of sustained

[1]Veterinary Epidemiology, Economics and Public Health Group, WOAH Collaborating Centre for Risk Analysis and Modelling, Department of Pathobiology and Population Sciences, The Royal Veterinary College, Hatfield, UK. [2]Department of Infectious Disease Epidemiology and International Health, London School of Hygiene & Tropical Medicine, London, UK. [3]Department of Medical Epidemiology and Biostatistics, Karolinska Institutet, Stockholm, Sweden. [4]Faculty of Veterinary Medicine, Jordan University of Science and Technology, Irbid, Jordan. [5]Department of Paediatrics, School of Medicine, Saint Louis University, St Louis, MO, USA. [6]These authors contributed equally: Punam Mangtani, Javier Guitian. ✉e-mail: pholloway3@rvc.ac.uk

vaccination initiatives, seroprevalences among small ruminant and cattle populations remain high[19–22]. While population level seroprevalences in Jordan remain largely unexamined, previous studies have reported seroprevalences among sub-populations including children, refugees, individuals working with livestock, those consuming livestock products and among hospitalised pyrexic patients[17,23–26].

Due to sociocultural and religious perceptions, camel milk is widely consumed raw across the region, while milk from other livestock species is usually boiled[27,28]. As a result, camels (*camelus dromedarius*) pose a unique public health threat from milk-borne pathogens, including *Brucella*[29,30]. However, while the public health threat associated with brucellosis among other livestock species in the region is widely recognized[10,31–33], the potential zoonotic impact of dromedary camels remains largely unexamined – despite previous reports of small-scale outbreaks in the region traced to the consumption of raw milk from infected camels[28,34–36]. In addition, the lack of a commercially available *Brucella* vaccine licensed for use in camels means that vaccination is limited to off-label use only, and while practiced in some countries, is not widespread across the region (including Jordan where government-led *Brucella* livestock vaccination programmes do not include camels, and access by the private sector is restricted)[37].

This study therefore represents the first large-scale population-based study examining the potential zoonotic threat posed by *Brucella* in camels among a high-risk population in the Middle East[10]. To address key knowledge gaps, we conducted a cross-sectional study among livestock-owning communities and their camels in southern Jordan (Aqaba and Ma'an governorates). The aims of this study were to, i) estimate the seroprevalence of *Brucella* among the camel population, ii) estimate the seroprevalence of *Brucella* among members of livestock-owning households (camel owning and non-camel owning), iii) identify risk factors for *Brucella* seropositive status among camel populations, iv) identify risk factors for *Brucella* seropositive status among livestock-owning household members, and v) identify socioculturally appropriate control strategies. In this work we show daily consumption of raw camel milk as being associated with *Brucella*

seropositive status on multivariable analysis, highlighting the need for socioculturally appropriate control measures, with targeted interventions among the camel reservoir being crucial for effective control.

## Results
### Camels

We collected blood samples from 902 camels, of which 26 samples were of insufficient volume for testing, with 884 samples remaining representing 227 herds. Of these 227 herds, 131 herds (455 camels) were owned by households included in a parallel human study, where household members were also tested for evidence of *Brucella* seropositive status. Median herd size was 10 [interquartile range (IQR) 5–20], with a mean of 3.9 camels sampled per herd (Fig. 1). The median age of camels sampled was 5 years (interquartile range [IQR] 2-8 years). There were 33/884 (3.7%) camels positive on Rose Bengal Plate test (RBPT), of which 8/884 (0.9%) were positive on Compliment Fixation Test (CFT), 9/884 (1.0%) were positive on indirect ELISA; 10/884 (1.1%) being positive on either CFT or indirect ELISA.

In Ma'an and Aqaba respectively, there were 7/340 (2.1%) and 1/544 (0.2%) positive camels (positive on RBPT and confirmed by CFT, as per OIE guidelines)[37], with true seroprevalence (adjusted for test performance values) being 2.4% (95% CI 1.1–4.8) and 0.1% (95% CI 0.0–1.1) respectively; true seroprevalence being 1.0% (95% CI 0.5–2.1) across both regions. There were 7/227 (3.1%) herds with at least one seropositive camel sampled, of which 6/81 (7.4%) were in Ma'an and 1/146 (0.7%) were in Aqaba. Adjusted herd-level seroprevalence (adjusted for test performance values and herd sampling proportion) was 30.4% (95% CI 22.2–39.5) in Ma'an and 4.8% (95% CI 2.1–8.2) in Aqaba; adjusted herd level seroprevalence (also weighted for region) across both regions being 18.7% (95% CI 14.3–23.3) (Table 1).

Due to the low seroprevalence of positives on RBPT and CFT (8/884, 0.9%), and the singularity of subsequent multivariable logistic regression models created, samples positive on RBPT and CFT or ELISA (10/884, 1.1%) were considered positive for evidence of *Brucella* exposure for purposes of logistic regression analyses to determine

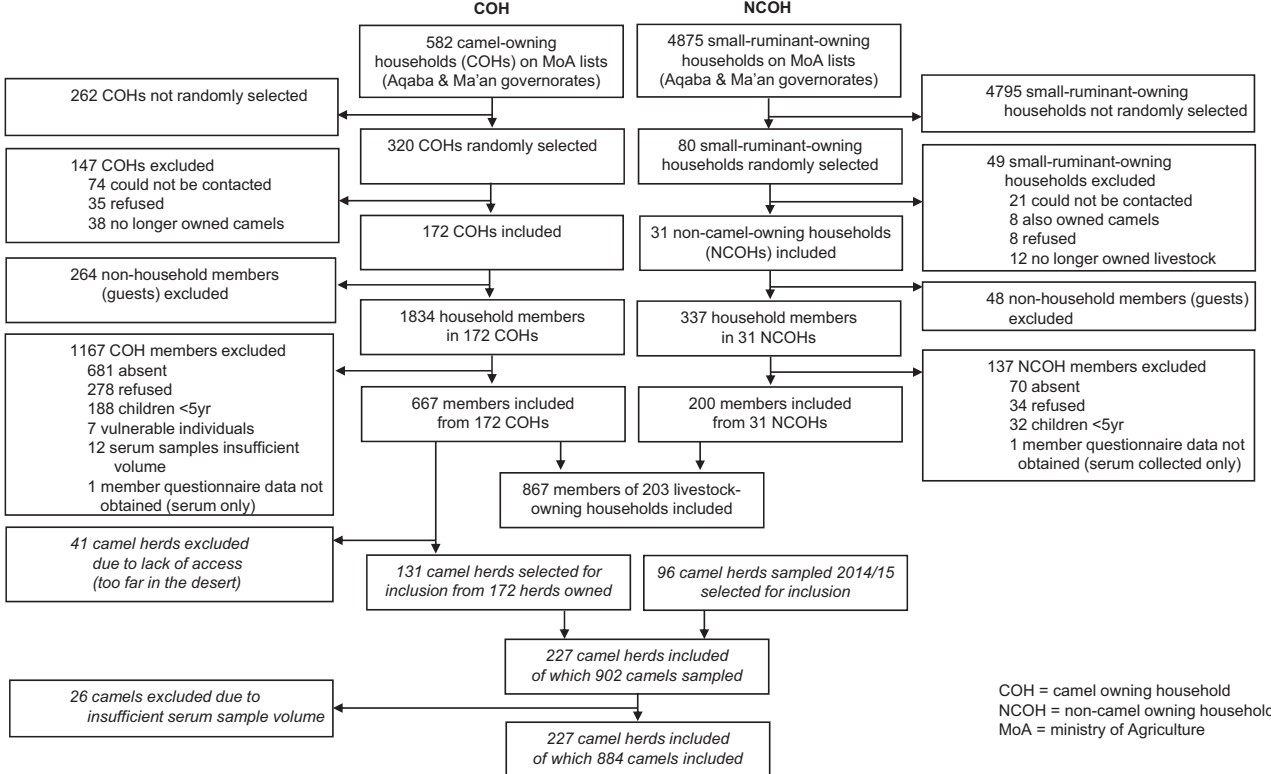

**Fig. 1 | Study profile.**

**Table 1 | Camel and herd-level *Brucella* seroprevalence among camel populations in southern Jordan**

| | Population seroprevalence | | | | Herd-level seroprevalence | | | |
|---|---|---|---|---|---|---|---|---|
| | Apparent prevalence | | True prevalence | | Unadjusted | | Adjusted* | |
| | Positive camels† / total camels | % | % | (95% CI) | Positive herds‡ / total herds | % | % | (95% CI) |
| Southern Jordan (Ma'an & Aqaba) | 8/884 | (0.9%) | 1.0% | (0.5–2.1) | 7/227 | 3.1% | 18.7% | (14.3–23.3) |
| Ma'an | 7/340 | (2.1%) | 2.4% | (1.1–4.8) | 6/81 | 7.4% | 30.4% | (22.2–39.5) |
| Aqaba | 1/544 | (0.2%) | 0.1% | (0.0–1.1) | 1/146 | 0.7% | 4.8% | (2.1–8.2) |

* Adjusted for combined test performance values; southern Jordan (Ma'an & Aqaba) estimate also weighted for region.
† Positive on RBPT and CFT.
‡ Herd contains one or more positive camels, positive on RBPT and CFT.

potential risk factors for infection. On univariable analysis, the following variables were found to be associated ($p < 0.20$) with seropositive status, region, purchasing camels (in the last year), closed herd management practices and one or more goats also being owned (Table 2). On multivariable analysis, positive status was associated with region (Ma'an) $OR_{adj}$ 6.23 (95%CI 1.53–41.81) $p = 0.0093$) and purchasing camels ($OR_{adj}$ 3.84 (95%CI 1.04–18.09) $p = 0.043$); with evidence to suggest closed herd management practices are protective (OR 0.00, p = 0.072) in a singular model (singularity being due to the absence of any positive camels sampled from among closed herds) (Table 3).

## Humans

We collected blood samples from 879 members of livestock-owning households of which 12 samples were of insufficient volume for diagnostic testing, with 867 samples remaining representing 203 households (Table 4). Median household size was 10 [IQR 7–13]), with a mean of 4.2 members sampled per household. The median age of individuals sampled was 27 (IQR 15–44). Of the 203 households sampled, 172 were camel-owning households (COH) (medium household size 10 [IQR 7–14]) including 667 household members (median age 27 years [IQR 16–44]), with a mean of 3.8 members sampled per household. The remaining 31 households were non-camel owning households (NCOH) (medium household size 11 [IQR 9–13]), including 200 household members (median age 26 years [IQR 15–44], with a mean of 6.5 individuals sampled per household (Fig. 1). Among COH, there were 131/172 (76.2%) households from which one or more camels from the herd were also tested for *Brucella* serological status. Jordanian Ministry of Agriculture (MoA) records show 3089 livestock owning households in Ma'an, of which 317/3089 (10.3%) own camels, and 1359 livestock-owning households in Aqaba, of which 265/1389 (19.1%) own camels, with camel ownership being 582/4448 (13.1%) overall. No households included in the study reported owning cattle (with cattle almost entirely absent from the south of Jordan, due to the arid, desert environment).

There were 68/867 (7.8%) individuals seropositive on RBPT and 62/867 (7.2%) individuals seropositive on IgG ELISA, with 91/867 (10.5%) seropositive to either test. A Kappa coefficient of 0.57 (95% CI 0.47–0.68) indicated moderate agreement between test results, individuals being considered positive if positive to either test[38]. The estimated intracluster correlation coefficient (ICC) for within household seropositivity to either test was 0.46 (95% CI 0.33–0.90). Adjusted population prevalence among members of livestock-owning households (adjusted for combined test performance values and weighted for region and camel ownership status) was 8.7% (95% CI 6.9–10.7). In Ma'an and Aqaba regions, 75/478 (15.7%) and 16/389 (4.1%) individuals were seropositive respectively, adjusted seroprevalences being 10.0% (95% CI 7.5–12.9) and 5.9% (95% CI 3.9–8.6) respectively (adjusted for combined test performance values and camel ownership status) (Table 5).

Among COH, 79/667 (11.8%) members were seropositive, adjusted seroprevalence (adjusted for combined test performance values and weighted for region) being 12.7% (95% CI 10.3–15.4). In Ma'an and Aqaba regions respectively, there were 68/379 (17.9%) and 11/288 (3.8%) seropositive individuals in COH; adjusted seroprevalences being 19.5% (95% CI 15.7–23.6) and 4.8% (95% CI 2.7–7.7) respectively. Among NCOH, 12/200 (6.0%) members were seropositive, with adjusted seroprevalence being 8.2% (95% CI 4.9–12.5); there were 7/99 (7.1%) and 5/101 (5.0%) positive individuals in Ma'an and Aqaba respectively, adjusted seroprevalences being 9.0% (95% CI 4.4–15.7) and 6.2% (95% CI 2.6–12.0) respectively (Table 5).

The following variables, with <10% missing values, were found to be associated (p <0.20) with seropositivity on RBPT or ELISA on univariable analysis: region, sub-region, region with camel owning status, sex, history of brucellosis, household dwelling type (tent), handwashing (>5 times/day), household owns sheep, drinking raw camel milk, drinking raw small ruminant milk, consuming raw small ruminant dairy products, birthing camels, birthing small ruminants, disposing of camel afterbirth, disposing of small ruminant afterbirth, slaughtering camels, slaughtering small ruminants, frequent interaction with livestock, any livestock engagement and hand hygiene when working with livestock (Table 6).

Pairwise correlations (R ≥0.4) were identified between birthing camels, disposing of camel afterbirth and slaughtering camels, with birthing of camels selected for inclusion in multivariable models. Pairwise correlations were also identified between frequently (≥weekly) working with livestock, frequently working with small ruminants, frequently working with camels and any history of livestock engagement, with frequently working with livestock selected for inclusion. Region, sub-region and region with camel owning status were also pairwise correlated, with region selected for inclusion. In addition, correlation was also identified between birthing small ruminants and disposing of small ruminant afterbirth, with birthing small ruminants selected for inclusion in multivariable models. Drinking raw milk from small ruminants and consuming raw small ruminant dairy products were correlated, with each variable tested in a separate multivariable model, adjusted for the same covariates. The composite variable hand hygiene when frequently working with livestock, was correlated with constituent variables frequently working with livestock and handwashing and therefore tested in a separate model adjusted for the same covariates.

On multivariable analysis, evidence of association was found between seropositivity and region (Ma'an) ($OR_{adj}$ 3.27 [1.86–6.07], p = <0.0001), drinking raw camel milk daily (categorical, p = 0.016), birthing small ruminants (≥weekly, in season) ($OR_{adj}$ 1.87 [1.02–3.39], p = 0.044), and poor hand hygiene (handwashing <5 times/day) when working with livestock (≥weekly) (categorical, p = 0.047) (Table 7).

Among individuals drinking raw camel milk daily, 27.6% (43/156) were seropositive in Ma'an and 4.9% (3/61) were seropositive in Aqaba (OR 9.42 [2.36–62.70], p = 0.0047), with daily consumption of raw camel milk associated with seropositive status in Ma'an, ($OR_{adj}$ 6.48 [3.22–13.74]) though not in Aqaba ($OR_{adj}$ 1.04 [0.23–3.47]) in a post-hoc logistic regression model, with Aqaba (not drinking camel milk daily) as a baseline and household as a random effect, adjusted for the

**Table 2 | Descriptive and univariable statistics of *Brucella* seropositivity among 884 camels from 227 herds in southern Jordan, using logistic regression analysis, likelihood ratio test**

| Variable | Serological status* | | | | | |
|---|---|---|---|---|---|---|
| | Total (missing) | +ve | % | OR | 95%CI | p |
| | 884 camels sampled | | | | | |
| Study period | | | | | | |
| 2014-15 | 429 | 5 | 1.2% | 0.94 | 0.26–3.41 | 0.93 |
| 2017-18 | 455 | 5 | 1.1% | | | |
| Region | | | | | | |
| Ma'an | 340 | 8 | 2.4% | 6.53 | 1.62–43.44 | 0.0069 |
| Aqaba | 544 | 2 | 0.4% | | | |
| Sex | | | | | | |
| Female | 669 | 9 | 1.3% | 2.81 | 0.52–51.96 | 0.26 |
| Male | 207 (8) | 1 | 0.5% | | | |
| Age >median (5 yr) | | | | | | |
| Yes | 405 | 7 | 1.5% | 0.41 | 0.09–1.50 | 0.18 |
| No | 474 (5) | 3 | 0.7% | | | |
| Herd size >median (10) | | | | | | |
| Yes | 552 | 8 | 1.4% | 2.43 | 0.60–16.14 | 0.23 |
| No | 332 | 2 | 0.6% | | | |
| Number of camel herds nearby (within a 15–minute drive) >20 | | | | | | |
| Yes | 364 | 6 | 1.6% | 2.06 | 0.58–8.10 | 0.26 |
| No | 495 (25) | 4 | 0.8% | | | |
| Herd is kept together as single group throughout the year | | | | | | |
| Yes | 600 | 6 | 1.0% | 0.66 | 0.19–2.62 | 0.54 |
| No | 267 (17) | 4 | 1.5% | | | |
| Herd has contact with other local herds | | | | | | |
| Yes | 519 | 8 | 1.3% | 1.15 | 0.32–5.36 | 0.84 |
| No | 255 (110) | 3 | 1.2% | | | |
| Herd is moved to distant areas for seasonal grazing purposes (transhumance) | | | | | | |
| Yes | 367 | 4 | 1.1% | 0.90 | 0.23–3.18 | 0.87 |
| No | 497 (20) | 6 | 1.2% | | | |
| New camels are purchased | | | | | | |
| Yes | 318 | 7 | 2.2% | 4.10 | 1.13 – 19.11 | 0.032 |
| No | 549 (17) | 3 | 0.5% | | | |
| Camels are borrowed for breeding purposes | | | | | | |
| Yes | 449 | 7 | 1.6% | 2.19 | 0.60–10.22 | 0.24 |
| No | 418 (17) | 3 | 0.7% | | | |
| Closed herd | | | | | | |
| Yes | 142 | 0 | 0.0% | 0.00 | – | 0.056 |
| No | 713 (29) | 10 | 1.4% | | | |
| Sheep are also owned | | | | | | |
| Yes | 533 | 8 | 1.5% | 2.53 | 0.63–16.83 | 0.21 |
| No | 334 (17) | 2 | 0.6% | | | |
| Goats are also owned | | | | | | |
| Yes | 603 | 9 | 1.5% | 3.98 | 0.74–73.67 | 0.12 |
| No | 264 (17) | 1 | 0.4% | | | |
| Small ruminant flocks, where owned, have been vaccinated against Brucella (Rev 1 vaccine)† | | | | | | |
| Yes | 263 | 4 | 1.5% | 1.17 | 0.26–6.00 | 0.84 |
| No | 231 (383) | 3 | 1.3% | | | |
| Dogs are present near the herd | | | | | | |
| Yes | 648 | 9 | 1.4% | 3.07 | 0.57–56.79 | 0.22 |
| No | 219 (17) | 1 | 0.5% | | | |

*Positive on RBPT and positive on CFT or ELISA.
†Once, or more, in the past 5 years.

same covariates (age, sex, region, household dwelling type, owning sheep, drinking camel milk, consuming small ruminant dairy products, birthing small ruminants and birthing camels). Similarly, region with camel owning status was associated with seropositivity (Ma'an COH ($OR_{adj}$ 4.24 [2.20–8.88]) with Aqaba COH as a baseline), when adjusted for the same covariates. Among COH where at least one camel was tested for *Brucella* seropositive status, seroprevalence among individuals drinking raw milk from a herd with a positive camel tested was 50.0% (2/4), compared to 15.7% (44/281) among households where camels tested in the herd were negative (OR 24.31 [0.69–851.92], p = 0.079). In a follow-up survey conducted among 69/172 (40.1%) of COH including 503 household members, 308/503 (61.2%) reported history of consuming camel milk in the past 6 months, of whom 285/308 (92.5%) reported always consuming this raw. Among 320/503 (63.6%) who reported history of consuming small ruminant milk in the past 6 months, 278/320 (86.9%) reported they never or rarely consumed this raw.

Among 577 individuals working frequently (≥weekly) with livestock, hand washing >5 times/day was found to be protective ($OR_{adj}$ 0.52 [0.30–0.92], p = 0.023), in a logistic regression model with household as a random effect adjusted for the same covariates (age, sex, region, household dwelling type, owning sheep, drinking camel milk, consuming small ruminant dairy products, birthing small ruminants and birthing camels). Use of soap (yesterday) when handwashing >5 times/day was protective ($OR_{adj}$ 0.52 [0.29–0.91]).

Among households owning small ruminants and reporting *Brucella* vaccination status with Rev 1 vaccine administered by the MoA (at least once during the previous 5 years), vaccine uptake was 65.4% (44/55) among flocks in Ma'an and 70.5% (31/44) among flocks in Aqaba, and 75.8% (75/99) overall. When weighted for camel owning status, estimated vaccine uptake among the study population was 89.0% in Ma'an and 87.4% in Aqaba, and 88.5% overall (vaccine uptake among COH flocks being 72.2% (57/79) compared to 90.0% (18/20) among NCOH flocks in the sample population, OR 0.29 [0.43–1.21]) p = 0.097). Vaccination of small ruminant flocks was found to be protective among household members owning vaccinated flocks; 17.8% (19/107) of individuals being seropositive among households owning unvaccinated flocks, compared to 8.8% (33/373) of individuals among households owning vaccinated flocks ($OR_{min.adj}$ 0.36 [0.14–0.91], p = 0.030), adjusted for a priori variables (age and sex) and region.

Among children of school age (≤15 yrs), seropositivity among those not attending school was 18.6% (8/43), compared to 6.3% (11/175) among those attending school; not attending school being associated with seropositive status ($OR_{min.adj}$ 3.23 [1.10–9.61], p = 0.033), when adjusted for priori variables (age and sex) and region. Among adults (>15 yrs), secondary education was found to be protective ($OR_{min.adj}$ 0.37 [0.18–0.79], p = 0.010), when adjusted for the same covariates. Among *Brucella* seropositive individuals included in the study, 74.7% (68/91) had no known history of a brucellosis.

## Discussion

Historically, dromedary camels have provided the only means of survival, transport, and sustained existence in the harsh desert environments of the Middle East and North Africa[39,40]. Consequently, they occupy a unique status within Arab Islamic culture[41]. The Qur'an (surra 88) entreats unbelievers to consider four things that God has made: the earth, the heavens, the mountains, and the camel – with camels perceived as being an example of God's power; a miracle of creation holding an entirely different position to all other animals[41]. The Islamic Hadith (the life and teachings of the prophet, transcribed by his followers), describes the healing powers of drinking camels' milk and urine[41–43]. As a result, camels are perceived as a uniquely clean livestock

**Table 3 | Multivariable associations between potential risk factors and *Brucella* seropositivity among camels from 227 herds in southern Jordan using logistic regression analysis, likelihood ratio test**

| Variable** | Category | A–priori adjusted OR (95% CI)[1] | | p value | Fully adjusted OR (95% CI)[2] | | p value |
|---|---|---|---|---|---|---|---|
| Region | Ma'an | 7.11 | 1.75 – 47.55 | 0.0050 | 6.23 | 1.53–41.81 | 0.0093 |
| Age | >median (5 yr) | 0.41 | 0.09 – 1.50 | 0.18 | 0.35 | 0.074–1.32 | 0.12 |
| Sex | Female | 3.56 | 0.64 – 66.50 | 0.17 | 3.86 | 0.68–72.73 | 0.14 |
| Camels are purchased | Yes | 4.32 | 1.19 – 20.20 | 0.026 | 3.84 | 1.04–18.09 | 0.043 |
| Closed herd[†] | Yes | 0.00 | NE | 0.047 | 0.00 | NE | 0.072 |

[1] Adjusted for a–priori variables: age, sex.
[2] Adjusted for a–priori variables age and sex, region and camels are purchased.
[*] 854 observations.
[†] In the last year.
[‡] Singular variable, adjusted for region, age, sex and camels are purchased, 842 observations.
*NE* non-estimable.

species among Islamic communities in the Middle East and beyond[44]. Study participants expressed the perception that owning camels was something of great honor – camels being seen as uniquely 'righteous' and 'clean' animals, compared to other livestock species[27]. A perception frequently expressed among the study population was that drinking camel milk imparted vital medicinal benefits, including increasing male libido, controlling blood sugar levels, healing gastrointestinal problems, and treating of cancer, amongst other medicinal properties[27,45]. Among the study population children drink camel's milk raw from an early age, with participants explaining that milk comes from the camel 'pure and ready to drink' – and should never be boiled, for fear of damaging its healing properties[26,46]. This perception, combined with an unfavourable change in taste after boiling, means that camel milk is almost universally consumed raw among camel-owning communities across the region, while milk from small ruminants (and cattle where present) is usually boiled due to a perception of risk from zoonotic diseases, such as *alhumaa almalitia* (brucellosis)[30,34,47].

It is within this context that our study findings identified daily consumption of raw camel milk as being associated with Brucella seropositive status among the study population OR$_{adj}$ 2.19 (95%CI 1.23–3.94), following multivariable analysis to adjust for potential confounders. When stratified by region, daily consumption of raw camel milk was found to be associated with seropositive status in Ma'an, though not in Aqaba. This reflects the high herd-level seroprevalences identified in Ma'an, where almost a third of herds were estimated to be positive (containing one or more seropositive camels), compared to Aqaba, where less than one in twenty herds were estimated to be positive. Likewise, half of all households members owning infected camels and drinking raw camel milk were found to be seropositive, compared to 15% among those drinking camel milk from herds where all sampled camels were seronegative. Even where individual camel seroprevalences may be relatively low (2.4% among camel populations in Ma'an, for example), widespread consumption of camel milk raw means that exposure to an infected animal becomes increasingly likely over time, particularly when drinking daily[34]. As such, boiling camel milk prior to consumption would have an important protective effect, particularly in areas where camel seroprevalences are higher (such as Ma'an)[46,48]. However, due to the strong sociocultural influences described (and an unfavourable change in taste after boiling) efforts to promote such profound behavioural change are likely to be challenging[46,48,49].

For this reason, control of *Brucella* among the camel reservoir (alongside other measures) is of paramount importance[50,51]. Following multivariable analysis, purchasing of camels (in the past year) was found to be associated with seropositive status among animals in the receiving herds (OR$_{adj}$ 3.84; 95%CI: 1.04–18.09). Due to their high economic value camels that abort are often sold at market, rather than slaughtered, with potential purchasers then unaware of their reproductive history and potential *Brucella* positive status (as reported by study participants)[22]. Conversely, study findings suggested evidence that closed herd management practices are protective against *Brucella* entering the herd (p = 0.072), in a singular model (all camels sampled in closed herds testing negative)[48]. To reduce potential within herd transmission, breeding camels should be separated from the main herd during parturition, as previously described[52,53]. To reduce potential transmission between livestock species, camel herds should be managed away from small ruminants where possible, particularly during lambing/kidding periods, as previously described[12,36,54]. Small ruminant seroprevalences in the region are known to be high[22], with Rev 1 (*B. melitensis*) vaccination of flocks offering a potentially protective effect, where camel herds and small ruminant flocks are managed together[50,51].

Camel herds in the study population had not been vaccinated against *Brucella*. Government-led vaccination programmes in Jordan do not include camels and private access to the Rev 1 vaccine is heavily restricted, with those herd owners surveyed reporting absence of vaccination in their herds. In addition, importing of camels from neighbouring countries, including Saudi Arabia (where Rev 1 vaccination may potentially have occurred), is prohibited unless for direct slaughter, suggesting the *Brucella* seroconversion identified in the study population is likely attributable to natural infection[44,50]. While off-label use of the live-attenuated Rev 1 vaccine in camels should be considered where appropriate (see FAO guidelines), challenges with efficacy and the risk of contaminated milk mean that protective herd management strategies likely offer the most accessible means of control for many herd owners (until licensed vaccines become available)[51]. Testing and slaughter schemes, using serum PCR testing where possible, including breeding males shared between herds, also offer a potential route toward reducing the risk of *Brucella* infection from raw camel milk in Jordan and the wider region (though such schemes require sufficient financial compensation be provided to succeed)[55].

Rev 1, being a live-attenuated vaccine, incurs risk of infection for veterinarians when administering the vaccine, and risk of abortion among pregnant livestock if accidently inoculated – both factors creating challenges in vaccine deployment across the region[56,57]. Reported Rev 1 vaccine uptake (at least once during the past 5 years) among small ruminant flocks was 89.0% in Ma'an and 87.4% among flocks in Aqaba, and 88.5% overall (weighted for camel-owning status), suggesting the observed difference in seroprevalence between regions is not attributable to differences in vaccine uptake. Vaccination of small ruminant flocks was found to offer a significantly protective effect among household members, with an apparent seroprevalence of 17.8% (19/107) among individuals owning unvaccinated flocks, compared to 8.8% (33/373) among those owning vaccinated flocks,

**Table 4 | Descriptive statistics of included members of livestock-owning households in southern Jordan, camel-owning (COH) and non-camel owning (NCOH) households**

| | COH (n = 667) | NCOH (n = 200) | All (n = 867) |
|---|---|---|---|
| **Age*** | | | |
| 5–14 | 143 (21.5%) | 47 (24%) | 190 (22%) |
| 15–24 | 151 (22.7%) | 50 (25%) | 201 (23%) |
| 25–39 | 161 (24.2%) | 40 (20%) | 201 (23%) |
| 40–59 | 137 (20.6%) | 44 (22%) | 181 (21%) |
| ≥60 | 74 (11.1%) | 18 (9%) | 92 (11%) |
| Median (IQR) | 27 (16–45) | 26 (15–44) | 27 (16–44) |
| **Sex** | | | |
| Female | 384 (57.6%) | 118 (59%) | 401 (46%) |
| Male | 283 (42.4%) | 82 (41%) | 466 (54%) |
| **Nationality††** | | | |
| Jordanian | 565 (93.1%) | 192 (99%) | 757 (95%) |
| Saudi Arabian | 23 (3.8%) | 0 (0%) | 23 (3%) |
| Egyptian | 8 (1.3%) | 0 (0%) | 8 (1%) |
| Sudanese | 8 (1.3%) | 0 (0%) | 8 (1%) |
| Other | 3 (0.5%) | 2 (1%) | 5 (1%) |
| **History of brucellosis** | | | |
| Yes | 18 (2.8%) | 5 (2.6%) | 23 (3%) |
| No | 629 (97.2%) | 189 (97.4%) | 818 (97%) |
| **Secondary education††** | | | |
| Yes | 180 (35.6%) | 53 (37%) | 233 (36%) |
| No | 326 (64.4%) | 90 (63%) | 414 (64%) |
| **Household size (no. of members >10)** | | | |
| 0–10 | 287 (43.0%) | 76 (38.0%) | 363 (42%) |
| 11–20 | 309 (46.3%) | 109 (54.5%) | 418 (48%) |
| >20 | 71 (10.6%) | 15 (7.5%) | 86 (10%) |
| **Household owns camels** | | | |
| Yes | 667 (100%) | 0 (0%) | 667 (77%) |
| No | 0 (0%) | 200 (100%) | 200 (23%) |
| **Household owns cattle** | | | |
| Yes | 0 (0%) | 0 (0%) | 0 (0%) |
| No | 667 (100%) | 200 (100%) | 867 (100%) |
| **Household owns small ruminants** | | | |
| Yes | 590 (88.5%) | 200 (100%) | 790 (91.1%) |
| No | 77 (11.5%) | 0 (0%) | 77 (8.9%) |
| **Where owned, small ruminants are reported as being vaccinated** | | | |
| yes | 255 (72.9%) | 118 (90.8%) | 373 (77.7%) |
| No | 95 (27.1%) | 12 (9.2%) | 107 (22.3%) |
| **Any history of livestock engagement†§** | | | |
| Yes | 571 (85.6%) | 711 (82.0%) | 711 (82%) |
| No | 96 (14.4%) | 156 (18.0%) | 156 (18%) |
| **Frequent (≥weekly) working with livestock¶** | | | |
| Yes | 478 (71.7%) | 577 (66.6%) | 577 (67%) |
| No | 189 (28.3%) | 290 (33.4%) | 290 (33%) |
| **Household home has an air-cooling system†** | | | |
| Yes | 169 (42.9%) | 56 (31.6%) | 225 (39.4%) |
| No | 225 (57.1%) | 121 (68.4%) | 346 (60.6%) |

*Missing values: Age n = 2 (COH n = 1, NCOH n = 1), Nationality n = 68 (COH n = 62, NCOH n = 6), Secondary education n = 219 (COH n = 162, NCOH n = 57). †Other nationalities included Ugandan (n = 1), Syrian (n = 1), British (n = 1). †Socioeconomic status indicator, due to sociocultural reasons socioeconomic status was not directly measured. §Any history of livestock activity (>never). ¶Involved in any recorded livestock activity weekly or more.

($OR_{min.adj}$ 0.36 [0.14–0.91]), when adjusted for a priori variables (age and sex) and region. These findings highlight the importance of small ruminant vaccination in protecting high-risk communities. Vaccine uptake was lower among COH (72.2%) compared to NCOH (90.0%), potentially explained by increased challenges in geographic accessibility to COH, compared to NCOH, and suggesting targeted interventions aimed at increasing Rev 1 uptake among small ruminant flocks owned by COH would have beneficial effects in reducing human rates of infection among such communities[22].

Birthing of small ruminants (≥weekly in season) was associated with seropositive status on multivariable analysis, $OR_{adj}$ 1.86 [1.00–3.38], consistent with previous findings, with weak evidence that drinking raw small ruminant milk (daily) also presents risk of infection, $OR_{adj}$ 1.96 [0.95–3.93]. These findings highlight the need for educational messages that reaffirm the importance of boiling small ruminant milk and dairy products before consumption[22,58], (due to its colloidal structure and protein composition, camel milk is unsuitable for making dairy products), and the use of gloves when birthing, where possible[59].

Infected livestock shed high volumes of *Brucella* in aborted materials and uterine discharges, with subsequent zoonotic transmission occurring via several potential routes, including ingestion of contaminated material on dirty hands and exposure of skin wounds (particularly on the hands and arms) or mucus membranes (eyes, nose and mouth) to contaminated materials[6]. Within this context, poor hand hygiene was found to be associated with *Brucella* seropositive status among individuals frequently working with livestock, with good hygiene (handwashing >5 times/day with soap) found to be protective ($OR_{adj}$ 0.52 [0.29–0.91]). Hand hygiene initiatives, such as WASH, therefore offer a potentially important control strategy for brucellosis among high risk communities in Jordan and the wider region[60]. (Due to frequency of handwashing being closely connected to concepts of prayer and ritual washing ('wudu') among Islamic communities, baseline handwashing frequency was set at ≤5 /day)[60].

As a WHO neglected disease, brucellosis presents a disease of poverty, with secondary education found to be protective among the study population ($OR_{min.adj}$ 0.37 [0.18–0.79]), while children not in school were at greater risk of infection, compared to contemporaries in school ($OR_{min.adj}$ 3.23 [1.10–9.61])[26,46]. Diagnosis rates among the study population were low, with just a quarter of seropositive individuals reporting a known history of brucellosis, suggesting a need for promotion of awareness regarding symptoms, diagnosis and treatment among rural communities in Jordan (and potentially the wider region), together with a need for increased capacity in diagnostic and medical services among such communities[58].

Among members of livestock-owning households in southern Jordan, *Brucella* seroprevalence was estimated to be 8.7% (95% CI 7.0-10.7) overall (adjusted for test performance values and weighted for region and camel ownership status), with adjusted seroprevalences in Ma'an and Aqaba being 10.0% (95% CI 7.5–12.9) and 5.9% (95% CI 3.9–8.6) respectively. Among COH, adjusted seroprevalences were 19.5% (95% CI 15.7–23.6) and 4.8% (95% CI 2.7–7.7) in Ma'an and Aqaba respectively, reflecting the high camel-herd level seroprevalences identified in Ma'an, compared to Aqaba (30.4% (22.2–39.5) and 4.8% (2.1–8.2) respectively). The ICC among included households was 0.46 (95% CI 0.33–0.90), suggesting moderate clustering; likely reflecting households owning infected herds (or flocks) with subsequent zoonotic transmission via consumption of infected milk (or milk products), alongside other potential routes.

This study has some limitations, with non-probabilistic and probabilistic sampling methods being used among camel herds during two distinct periods (2014–15 and 2017–18 respectively), however, we do not believe this has had an important impact on seroprevalence estimates or risk factor analysis (with seroprevalence among randomly

**Table 5 | Individual *Brucella* seroprevalence among livestock-owning household members in southern Jordan**

| | | Unadjusted | Adjusted* | | |
|---|---|---|---|---|---|
| | | Positive individuals / total individuals | % | % | (95% CI) |
| Southern Jordan (Ma'an & Aqaba) | COH[†] | 79/667 | (11.8%) | 12.7% | (10.3–15.4) |
| | NCOH[‡] | 12/200 | (6.0%) | 8.2% | (4.9–12.5) |
| | All households | 91/867 | (10.5%) | 8.7% | (6.9–10.7) |
| Ma'an | COH[†] | 68/379 | (17.9%) | 19.5% | (15.7–23.6) |
| | NCOH[‡] | 7/99 | (7.1%) | 9.0% | (4.4–15.7) |
| | All households | 75/478 | (15.7%) | 10.0% | (7.5–12.9) |
| Aqaba | COH[†] | 11/288 | (3.8%) | 4.8% | (2.7–7.7) |
| | NCOH[‡] | 5/101 | (5.0%) | 6.2% | (2.6–12.0) |
| | All households | 16/389 | (4.1%) | 5.9% | (3.9–8.6) |

[†]Camel owning household.

[‡]Non-camel owning household (small ruminants only).

*Adjusted for combined test performance values, with Southern Jordan (Ma'an & Aqaba) estimates also weighted for region. All households estimates are also adjusted for camel owning status.

selected camel herds being similar to those among non-probabilistically sampled herds, across the study period).

In conclusion, Brucellosis represents a serious public health threat among livestock-owning communities in southern Jordan and the wider region, with camels presenting an important source of zoonotic transmission via consumption of raw milk (with milk from other livestock species being largely consumed boiled). Due to deep socio-cultural perceptions, efforts to encourage boiling of camel milk prior to consumption are likely to be challenging, (though should be attempted nonetheless, in a culturally appropriate context). For this reason, reducing *Brucella* infection rates among camel populations in Jordan, and the wider region, is of crucial importance in protecting human populations associated with such herds from infection via consumption of raw camel milk. This is potentially best achieved through a combination of targeted management practices (such as closed herd management, with purchasing only where necessary from trusted (preferably tested) sources), alongside possible government-led test and slaughter schemes and off-label use of the *B. melitensis* Rev 1 vaccine, where appropriate. Small ruminant vaccination had a significantly protective effect on associated household members, highlighting the importance of government-led vaccination programmes and the need for increased capacity, so as to improve vaccine uptake where possible (a quarter of flocks in sample population had not being vaccinated during the previous 5 years). Among individuals working frequently with livestock (≥weekly) handwashing with soap >5 times/day was found to be significantly protective, highlighting the importance of initiatives that promote hand hygiene, such as WASH, among high-risk rural communities in the region. Education had a significantly protective effect (including through being in school), highlighting the importance of education promotion strategies and increased capacity (including accessibility) among high-risk rural communities in the region (with challenges to this for desert dwelling, semi-nomadic Bedouin communities, such as the study population). Levels of diagnosis were low, suggesting the need for improved awareness strategies and increased capacity for diagnostic testing and treatment among high-risk rural communities in the region.

## Methods
### Ethical approval
The present study was done with institutional ethical review board approvals from The Royal Veterinary College and London School of Hygiene & Tropical Medicine (both London, UK) and Jordan University of Science and Technology (JUST) (Irbid, Jordan).

### Camels
**Study design and participants.** Camel sampling was conducted in Aqaba and Ma'an governorates of southern Jordan across two distinct periods during 2014–2015 and 2017–2018 (Fig. 2). Due to the absence of an adequate sampling frame between February 9th 2014 and December 30th 2015, we conducted non-probabilistic sampling of herds belonging to camel-owing clients of a centrally located private veterinary practice (Al Quwayrah, Aqaba governorate) and camel-owing clients of local government veterinarians working in the study area. Between September 27th, 2017, and October 11th, 2018, we conducted a cross-sectional study using a sampling frame provided by the MoA comprising a list of all households owning one or more camels within the four sub-regions (Aqaba East, Aqaba West, Ma'an East, and Ma'an West), with herds being randomly selected by a co-author (SN) using computer-generated randomisation lists (Stata, version 15.1); (these camel-owing households being also eligible for inclusion in the parallel human study).

Institutional and national guidelines for the care of animals were followed at all times. Informed consent was obtained from all herd owners at the time of sampling, with a veterinary surgeon clinically examining all camels included in the study before sampling.

**Sample collection and laboratory procedures.** Selected herds were visited by members of the sampling team, including a veterinarian and community facilitator, who provided study information, assessed eligibility, and obtained consent. Based on expected owner compliance, in herds >12 camels we sampled 12 camels per herd and in herds of ≤12, we sampled all camels, subject to accessibility and owner permissions. Blood samples were collected from selected camels alongside administration of a structured questionnaire regarding potential camel-level and herd-level risk factors for *Brucella* exposure during the previous year, including *Brucella* vaccination. Questionnaires were administered in the local dialect on paper (2014–2015) and on android tablets (2017–2018) using the application Open Data Kit (version 1.10). Blood samples were collected in 8 mL plain serum vacutainer tubes and centrifuged centrally in Aqaba, at 2000 revolutions per minute (961×g) for 10 min, followed by serum collection and storage at −20 °C. All samples were stored centrally in Aqaba before transport to the diagnostic laboratory at the Veterinary Health Centre, JUST, Irbid for laboratory testing.

Serum samples were tested using the Rose-Bengal test (RBPT) for the detection of antibodies against *Brucella* antigens supplied by Jovac, Amman, Jordan. Briefly, a sample of serum (0.05 ml) was mixed with

**Table 6 | Descriptive and univariate statistics of *Brucella* seropositivity among 867 members of 203 livestock owning households in southern Jordan using logistic regression analysis with household as a random effect, likelihood ratio test**

| Category* | Total 867 (missing) | +ve | % | OR | 95% CI | p |
|---|---|---|---|---|---|---|
| **Spatial data** | | | | | | |
| Region | | | | | | |
| Ma'an | 478 | 75 | 15.7% | 5.32 | 2.42–11.67 | <0.0001 |
| Aqaba | 389 | 16 | 4.1% | 1.00 | | |
| Sub-region | | | | | | |
| Aqaba West | 186 | 5 | 2.7% | 1.00 | | 0.0002 |
| Aqaba East | 203 | 11 | 5.4% | 2.06 | 0.55–7.70 | |
| Ma'an East | 222 | 27 | 12.2% | 6.14 | 1.80–20.94 | |
| Ma'an West | 256 | 48 | 18.8% | 9.64 | 2.96–31.45 | |
| Region with camel owning status | | | | | | |
| Aqaba COH | 288 | 11 | 3.8% | 1.00 | | 0.0001 |
| Aqaba NCOH | 101 | 5 | 5.0% | 1.29 | 0.27–5.46 | |
| Ma'an COH | 379 | 68 | 17.9% | 6.57 | 2.91–16.85 | |
| Ma'an NCOH | 99 | 7 | 7.1% | 2.17 | 0.58–8.11 | |
| **Personal information** | | | | | | |
| Sex | | | | | | |
| Female | 401 | 34 | 8.5% | 0.70 | 0.42–1.18 | 0.18 |
| Male | 466 | 57 | 12.2% | 1.00 | | |
| Age | | | | | | |
| 5–15 | 190 (2) | 17 | 8.9% | 1.00 | | 0.39 |
| >15–25 | 201 | 30 | 14.9% | 2.14 | 1.01–4.56 | |
| >25–40 | 201 | 17 | 8.5% | 1.45 | 0.62–3.38 | |
| >40–60 | 181 | 19 | 10.5% | 1.62 | 0.72–3.66 | |
| >60 | 92 | 8 | 8.7% | 1.35 | 0.47–3.84 | |
| Nationality | | | | | | |
| Jordanian | 757 (66) | 81 | 10.7% | 1.00 | NE | 0.62 |
| Egyptian | 8 | 2 | 25.0% | 6.17 | | |
| Sudanese | 8 | 0 | 0.0% | 0.00 | | |
| Saudi Arabian | 23 | 0 | 0.0% | 0.00 | | |
| Other | 5 | 0 | 0.0% | 0.00 | | |
| Household employee | | | | | | |
| No | 832 | 87 | 10.5% | 1.00 | | 0.74 |
| Farm Worker | 23 | 2 | 8.7% | 1.11 | 0.19–6.40 | |
| Slaughterer | 12 | 2 | 16.7% | 2.37 | 0.26–21.28 | |
| Secondary education | | | | | | |
| Yes | 233 (218) | 17 | 7.3% | 0.47 | 0.24–0.93 | 0.029 |
| No | 416 | 55 | 13.2% | 1.00 | | |
| Currently in school (if ≤18 yrs of age) | | | | | | |
| Yes | 175 (649) | 11 | 6.3% | 0.32 | 0.11–0.92 | 0.035 |
| No | 43 | 8 | 18.6% | 1.00 | | |
| **Household information** | | | | | | |
| Household dwelling type (tent) | | | | | | |
| Yes | 296 | 47 | 15.9% | 2.29 | 1.11–4.74 | 0.025 |
| No | 571 | 44 | 7.7% | 1.00 | | |
| Household size | | | | | | |
| 1 -10 | 363 | 36 | 9.9% | 1.00 | | 0.51 |
| 11-20 | 418 | 43 | 10.3% | 0.91 | 0.42–1.96 | |
| >20 | 86 | 12 | 14.0% | 2.05 | 0.52–8.05 | |

| Category* | Total 867 (missing) | +ve | % | OR | 95% CI | p |
|---|---|---|---|---|---|---|
| Household owns camels | | | | | | |
| 0 | 200 | 12 | 6.0% | 1.00 | | 0.33 |
| 1-6 | 267 | 29 | 10.9% | 1.73 | 0.58–5.16 | |
| >6 | 400 | 50 | 12.5% | 2.17 | 0.78–6.04 | |
| Household owns sheep | | | | | | |
| No | 248 | 17 | 6.9% | 1.00 | | 0.033 |
| ≤50 | 338 | 32 | 9.5% | 1.59 | 0.61–4.16 | |
| >50 | 281 | 42 | 14.9% | 3.54 | 1.32–9.53 | |
| Household owns goats | | | | | | |
| No | 118 | 13 | 11.0% | 1.00 | | 0.94 |
| ≤50 | 479 | 50 | 10.4% | 1.11 | 0.37–3.34 | |
| >50 | 270 | 28 | 10.4% | 1.24 | 0.37–4.14 | |
| Small ruminants are reported vaccinated against Brucella (where small ruminants owned) | | | | | | |
| Yes | 373 (387) | 33 | 8.8% | 0.45 | 0.17–1.18 | 0.10 |
| No | 107 | 19 | 17.8% | 1.00 | | |
| **Consumption of livestock products** | | | | | | |
| Drinking raw camels' milk | | | | | | |
| No | 474 | 32 | 6.8% | 1.00 | | 0.0001 |
| ≤weekly | 176 | 13 | 7.4% | 1.09 | 0.51–2.36 | |
| Daily | 217 | 46 | 21.2% | 4.01 | 2.05–7.82 | |
| Daily consumption of camel by region | | | | | | |
| Aqaba | 61 | 3 | 4.9% | 1.00 | 2.36–62.70 | 0.0047 |
| Ma'an | 156 | 43 | 27.6% | 9.42 | | |
| Drinking raw sheep or goat milk (owning sheep, owning goats) | | | | | | |
| No | 707 | 59 | 8.3% | 1.00 | | 0.012 |
| ≤weekly | 89 | 13 | 14.6% | 2.02 | 0.92–4.40 | |
| Daily | 71 | 19 | 26.8% | 2.97 | 1.33–6.60 | |
| Consuming raw dairy products made from small ruminant milk | | | | | | |
| No | 630 | 56 | 8.9% | 1.00 | | 0.054 |
| ≤weekly | 177 | 19 | 10.7% | 1.29 | 0.67–2.49 | |
| Daily | 60 | 16 | 26.7% | 2.80 | 1.21–6.48 | |
| **Livestock engagement activities** | | | | | | |
| Birthing camels (>never) | | | | | | |
| Yes | 299 | 44 | 14.7% | 1.70 | 0.98–2.94 | 0.058 |
| No | 568 | 47 | 8.3% | 1.00 | | |
| Birthing small ruminants (≥weekly in season) | | | | | | |
| Yes | 122 | 27 | 22.1% | 2.95 | 1.58–5.50 | 0.0007 |
| No | 745 | 64 | 8.6% | 1.00 | | |
| Disposing of camel afterbirth (>never) | | | | | | |
| Yes | 634 | 57 | 9.0% | 1.00 | | 0.11 |
| No | 233 | 34 | 14.6% | 1.60 | 0.90–.82 | |
| Disposing of small ruminant afterbirth (≥weekly in season) | | | | | | |
| Yes | 117 | 22 | 18.8% | 2.27 | 1.18–4.38 | 0.014 |
| No | 750 | 69 | 9.2% | 1.00 | | |
| Slaughtering small ruminants (≥weekly) | | | | | | |
| Yes | 82 | 15 | 18.3% | 3.02 | 1.36–6.70 | 0.0065 |
| No | 785 | 76 | 9.7% | 1.00 | | |

**Table 6 (continued) | Descriptive and univariate statistics of Brucella seropositivity among 867 members of 203 livestock owning households in southern Jordan using logistic regression analysis with household as a random effect, likelihood ratio test**

| Category* | Total 867 (missing) | +ve | % | OR | 95% CI | p |
|---|---|---|---|---|---|---|
| Frequently (≥weekly) working with camels | | | | | | |
| Yes | 439 | 61 | 13.9% | 1.81 | 0.99–3.28 | 0.053 |
| No | 428 | 30 | 7.0% | 1.00 | | |
| Frequently (≥weekly) working with small ruminants | | | | | | |
| Yes | 509 | 73 | 14.3% | 3.06 | 1.62–5.78 | 0.0006 |
| No | 358 | 18 | 5.0% | 1.00 | | |
| Frequently (≥weekly) working with livestock | | | | | | |
| Yes | 577 | 75 | 13.0% | 2.49 | 1.27–4.89 | 0.0080 |
| No | 290 | 16 | 5.5% | 1.00 | | |
| Any history livestock engagement | | | | | | |
| Yes | 711 | 83 | 11.7% | 2.09 | 0.87–4.99 | 0.099 |
| No | 156 | 8 | 5.1% | 1.00 | | |
| **Hand hygiene** | | | | | | |
| Hand washing >5 times / day | | | | | | |
| Yes | 528 | 42 | 8.0% | 0.54 | 0.32–0.91 | 0.021 |
| No | 339 | 49 | 14.5% | 1.00 | | |
| Used soap yesterday | | | | | | |
| Yes | 777 | 77 | 9.9% | 0.75 | 0.35–1.63 | 0.47 |
| No | 90 | 14 | 15.6% | 1.00 | | |
| Frequently (≥weekly) working with livestock, with handwashing frequency | | | | | | |
| No | 428 | 30 | 5.5% | 1.00 | | 0.0009 |
| Yes (>5 times/day) | 255 | 26 | 9.4% | 1.78 | 0.85–3.69 | |
| Yes (≤5 times/day) | 184 | 35 | 18.3% | 3.87 | 1.83–8.15 | |
| Among individuals frequently working with livestock, handwashing >5 times day | | | | | | |
| Yes | 342 (290) | 32 | 9.4% | 0.46 | 0.25–0.84 | 0.012 |
| No | 235 | 43 | 18.3% | 1.00 | | |
| Among individuals frequently working with livestock, use of soap yesterday | | | | | | |
| Yes | 504 (290) | 62 | 12.3% | 0.70 | 0.30–1.64 | 0.41 |
| No | 73 | 13 | 17.8% | 1.00 | | |
| Among individuals frequently working with livestock, handwashing with or without reported use of soap yesterday | | | | | | |
| ≤5 times/day | 235 (290) | 43 | 18.3% | 1.00 | | 0.036 |
| >5 times/day without soap | 26 | 3 | 11.5% | 0.70 | 0.15–3.28 | |
| >5 times/day with soap | 316 | 29 | 9.2% | 0.44 | 0.24–0.82 | |
| **Camel herd tested for brucella seropositivity** | | | | | | |
| A camel in the herd tested seropositive for Brucella (positive on RBPT, confirmed positive on CFT) | | | | | | |
| Yes | 7 | 2 | 28.6% | 1.00 | 0.36–139.38 | 0.20 |
| No | 489 | 56 | 11.5% | 7.10 | | |
| Drinking camel milk where a camel in the herd tested seropositive for Brucella (positive on RBPT, confirmed positive on CFT) | | | | | | |
| Yes | 4 | 2 | 50% | 1.00 | 0.69–851.92 | 0.079 |
| No | 281 | 44 | 16% | 24.31 | | |

0.05 ml of antigen on a microscope slide, producing a zone approximately 2 cm in diameter. This mixture was then gently agitated at room temperature for four minutes and observed for agglutination, the sample being considered positive where a visible reaction was observed. Sensitivity and specificity were assumed to be 86.7% and 99.1% respectively[61]. Serum samples positive on RBPT were also tested using Compliment Fixation Test (CFT) using standardized *Brucella* antigens supplied by Jovac, Amman, Jordan, as described by the OIE[37]. CFT plates were left to stand for one hour to allow unlysed cells to settle before results were read, with positive and negative controls being run for test validation. Results ≥20 IU/ml were considered positive, based on an absence of haemolysis. Sensitivity and specificity were estimated to be 99.0% and 98.4% respectively[62]. (Combined performance values of RBPT confirmed on CFT being 85.8% and >99.9% respectively).

Serum samples positive on RBPT were also tested using a commercially available multispecies *Brucella* IgG indirect ELISA supplied by Idvet (with these results being used for purposes of risk factor analysis, together with those samples positive on CFT)[63] Tests were performed according to the instructions of the manufacturer, with optical densities measured at 450 nm. For each sample, the %OD was obtained as: $\%OD = 100 \times (S–N)/(P–N)$, where S is the sample OD, N is the OD of the negative control and P the OD of the positive control. Samples with a %OD ≥120% were considered positive. Sensitivity and specificity were estimated as being 74.4% and 96.3% respectively[64–66].

**Outcomes.** The primary outcome was evidence of *Brucella* seropositive status (seropositive on RBPT, confirmed seropositive on CFT)[37]. Secondary outcomes were potential associations between possible risk factors and evidence of *Brucella* exposure (seropositive on RBPT, confirmed by seropositivity on either CFT or ELISA).

**Statistical analysis.** A sample size estimate of 903 camels was used, with a total camel population size of 7750 (MoA figures for Aqaba and Ma'an governorates), an expected seroprevalence of 12.1% (Al Majali et al.)[36], a ± 2% precision, 95% confidence interval and design effect of 1.0. Seroprevalence estimates and their 95% CIs were calculated at the camel and herd levels, with herd-level prevalence defined according to herds with at least one seropositive individual.

To adjust for herd sampling proportion, herd-level seroprevalence was estimated accounting for the uncertainty arising from sampling only a proportion of each herd (the proportion being different in each herd), based on the method described by Beauvais et al.[67]. Due to low prevalence, a binomial prior distribution for the number of positives in each herd was used, using a true prevalence distribution calculated for each region. This was used in a Bayesian computation with herd sample results, giving a discrete probability distribution for positives within a herd. Each herd was then simulated as being positive or negative using a random sample from a binomial distribution. This was repeated 1,000 times to create an uncertainty distribution, where the 2.5th and 97.5th percentiles gave a 95% credible interval and the 50th percentile gave the most likely herd-level prevalence.

Univariable associations between hypothetical risk factors and serological status were assessed using logistic regression, with serological status as a binary outcome, without weighting for region. Potential risk factors, including age, sex, racing status and herd management practices during the previous year, were analysed as categorical variables at the individual level (including the herd-level management practice variables).

Variables with a p value <0·20 in univariable analysis were considered for inclusion in the multivariable models[48]. Collinearities between variables were examined with use of the Pearson's coefficient, with a threshold of 0·4 or greater to define direct collinearity and −0·4 for inverse collinearity. Any collinear variables were to be excluded from the same multivariable model and tested in separate models, except for a-priori variables (age and sex), which were included alongside any collinear variables in the same model. Models that contained a singular variable were rebuilt with the singular variable

**Table 7 | Multivariable associations between potential risk factors and *Brucella* seropositivity among members of 203 livestock owning households in southern Jordan using logistic regression analysis with household as a random effect, likelihood ratio test**

| Variable[*¶] | Category | A–priori adjusted OR (95% CI)[1] | | p value | Fully adjusted OR (95% CI)[2] | | p value |
|---|---|---|---|---|---|---|---|
| Sex | Female | 0.71 | 0.42–1.18 | 0.19 | 0.86 | 0.51–1.45 | 0.57 |
| Age | 5–15 | 1.00 | | 0.25 | 1.00 | | 0.12 |
| | >15–25 | 2.32 | 1.09–4.94 | | 2.02 | 1.01–4.15 | |
| | >25–40 | 1.40 | 0.61–3.25 | | 0.96 | 0.44–2.08 | |
| | >40–60 | 1.78 | 0.79–4.01 | | 1.46 | 0.68–3.17 | |
| | >60 | 1.28 | 0.46–3.61 | | 0.88 | 0.33–2.20 | |
| Region | Ma'an | 5.51 | 2.53–12.03 | <0.0001 | 3.27 | 1.86–6.07 | <0.0001 |
| Primary dwelling tent | Yes | 2.25 | 1.15–4.42 | 0.018 | 1.52 | 0.92–2.51 | 0.10 |
| Household own sheep | No | 1.00 | | 0.065 | 1.00 | | 0.43 |
| | 1–50 | 1.85 | 0.73–4.66 | | 1.22 | 0.63–2.41 | |
| | >50 | 3.04 | 1.19–7.72 | | 1.52 | 0.80–2.98 | |
| Birthing camels | Yes | 1.45 | 0.80–2.62 | 0.22 | 0.88 | 0.50–1.54 | 0.65 |
| Birthing small ruminants | Yes | 2.45 | 1.31–4.58 | 0.0050 | 1.87 | 1.02–3.39 | 0.044 |
| Frequently (≥weekly) working with livestock | Yes | 2.02 | 1.02–4.03 | 0.045 | 1.30 | 0.68–2.58 | 0.43 |
| Handwashing ≤5 times daily | Yes | 0.57 | 0.34–0.98 | 0.041 | 0.64 | 0.39–1.04 | 0.069 |
| Drinking camel milk | Rarely/never | 1.00 | | 0.0012 | 1.00 | | 0.016 |
| | Weekly/monthly | 1.12 | 0.51–2.45 | | 1.02 | 0.48–2.06 | |
| | Daily | 3.26 | 1.66–6.41 | | 2.19 | 1.23–3.94 | |
| Consuming small ruminant dairy products | Rarely/never | 1.00 | | 0.088 | 1.00 | | 0.65 |
| | Weekly/monthly | 1.39 | 0.72–2.67 | | 1.11 | 0.60–1.99 | |
| | Daily | 2.46 | 1.08–5.63 | | 1.43 | 0.66–2.97 | |
| Drinking raw sheep or goat milk[†] | Rarely/never | 1.00 | | 0.0047 | 1.00 | | 0.071 |
| | Weekly/monthly | 2.32 | 1.07–5.05 | | 1.93 | 0.92–3.82 | |
| | Daily | 3.15 | 1.43–6.93 | | 1.96 | 0.95–3.93 | |
| Frequently working with livestock (≥weekly) and handwashing frequency[‡] | Not frequently working with livestock | 1.00 | | 0.0063 | 1.00 | | 0.047 |
| | Frequently working with livestock & hand washing >5 times/day | 1.45 | 0.68–3.07 | | 0.94 | 0.46–1.95 | |
| | Frequently working with livestock & hand washing ≤5 times/day | 3.03 | 1.42–6.47 | | 1.82 | 0.90–3.77 | |

[1] Adjusted for a–priori variables: age, sex.

[2] Adjusted for a–priori variables age, sex, region, household dwelling type, owning sheep, drinking camel milk, consuming small ruminant dairy products, birthing small ruminants and birthing camels.

[*] In the last 12 months (with exception of age and sex).

[†] Due to collinearity (Pearson R coefficient ≥0.4) with consumption of raw sheep or goat dairy products this variable was tested in separate multivariable model in place of consumption of raw sheep or goat dairy products. In this model drinking raw camel milk, birthing small ruminants and region continued to demonstrate significant association (p< 0.05) with *Brucella* seropositive status.

[‡] Due to collinearity (Pearson R coefficient ≥0.4) with the variables frequently working with livestock and handwashing >5 day, this variable was tested in separate multivariable model in place of frequently working with livestock and handwashing >5 day. In this model drinking raw camel milk, birthing small ruminants and region continued to demonstrate significant association (p < 0.05) with *Brucella* seropositive status.

[¶] 865 observations in all models.

excluded, with the singular variable then adjusted in a separate model using the same covariates.

Multivariable models were constructed using logistic regression, with herd as a random effect. Models were first constructed with a backwards stepwise method; the variable to be removed at each step was identified by comparing models with each of the non-a-priori variables removed and selecting the model with the lowest Akaike Information Criterion score, provided that removing the variable did not change the log odds ratio of any non-a-priori variables by more than 10%[48]. Models were then constructed with a forward stepwise method to ensure the same model was constructed by either method. Regression analysis confidence intervals were generated with the profile likelihood technique.

All statistical analyses were done in R (version 4.0.3).

### Humans

**Study design and participants.** We conducted a cross-sectional study between September 27th, and October 11th, 2018, among

livestock-owning households in four sub-regions of southern Jordan (Aqaba East, Aqaba West, Ma'an East, and Ma'an West) (Fig. 3). Livestock-owning households were randomly selected by a co-author (SN) using computer-generated randomisation lists (Stata, version 15.1) from a sampling frame provided by the MoA, comprising a list of all households owning one or more camels, and a list of those owning sheep or goats, within the four sub-regions. Both house and tent dwellings were eligible for inclusion.

All household members present at the time of sampling were invited to participate, except children younger than 5 years or individuals without the capacity to provide informed consent (as judged by a medical clinician and household head; the household head being the listed herd or flock owner in MoA lists). Household membership was defined as either: 1) self-identifying as a household member and sleeping in the household the previous night; or 2) being a household employee.

Written informed consent was obtained from all participating individuals at the time of sampling, along with parent or guardian

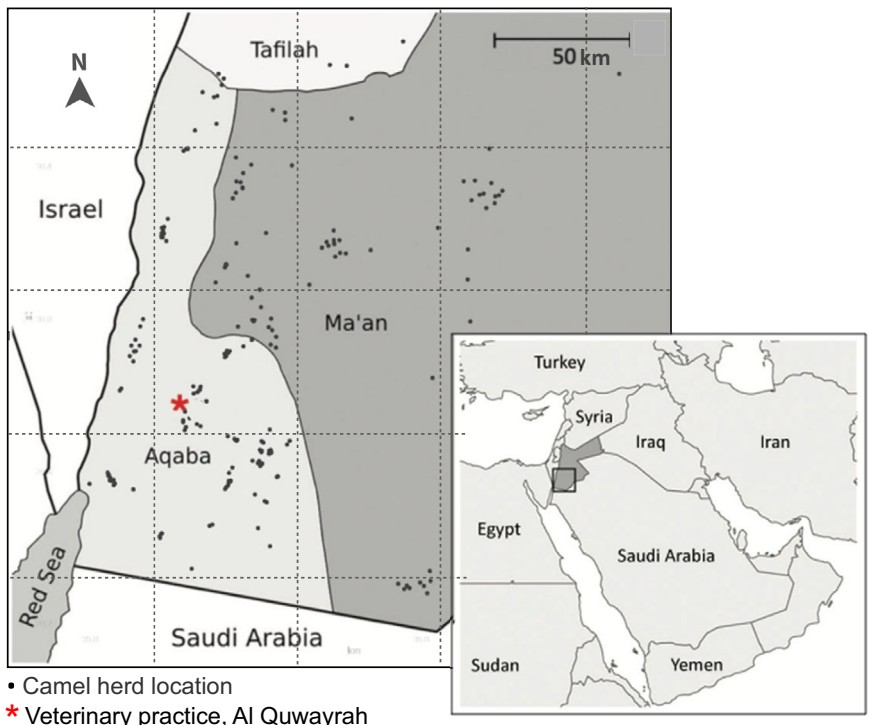

• Camel herd location
* Veterinary practice, Al Quwayrah

**Fig. 2 | Location of 227 camel herds sampled for *Brucella* in southern Jordan February 2014 to October 2018 (due to local grazing movements there were three herds selected from the MoA list for Ma'an (west) that were sampled in the neighbouring region, Tafilah. Results from these herds were attributed to Ma'an.).**

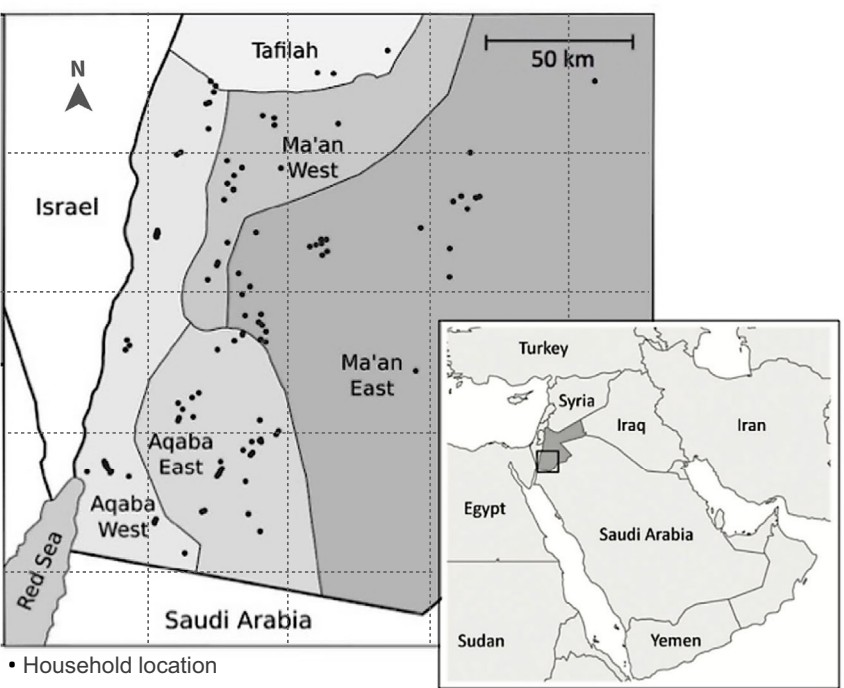

• Household location

**Fig. 3 | Location of 203 livestock-owning households sampled for *Brucella* in southern Jordan, September 2017 to October 2018.**

consent for children aged 5–15 years. Institutional and national guidelines for the care of participating individuals were followed at all times.

**Sample collection and laboratory procedures.** Selected households were visited by members of the sampling team, including a medical practitioner and community facilitator, who provided study

information, assessed eligibility, and obtained consent. Blood samples were collected from participating individuals together with administration of a structured questionnaire regarding potential risk factors for *Brucella* exposure during the previous 6 months, including questions regarding spatial data, personal information (including age and sex (observed), household information, consumption of livestock

products, livestock engagement activities and hand hygiene (with a follow-up questionnaire administered among COH regarding milk boiling practices). An additional questionnaire was administered to the household head regarding household-level risk factors, including age and sex of any unsampled members. Questionnaires were administered in the local dialect on android tablets using the application Open Data Kit (version 1.10). Due to cultural sensitivity, information regarding participant sex was obtained from interviewer observation. Information regarding nationality was self-reported, according to national identity card or passport details.

Blood samples were collected in 8 mL plain serum vacutainer tubes and centrifuged centrally in Aqaba, at 2000 revolutions per minute (961×g) for 10 min, followed by serum collection and storage at −20 °C. All samples were stored centrally in Aqaba before transport to the diagnostic laboratory at the Veterinary Health Centre, JUST, Irbid for laboratory testing.

Serum samples were tested for antibodies against *Brucella* using the commercially available *Brucella* IgG ELISA *classic,* supplied by Institut Virion/Serion, GmbH, Würzburg, Germany[68]. The technique was performed according to the manufacturer's instructions, with optical densities measured at 450 nm. Samples with a titre above 25 IU/ml were considered positive for IgG. Sensitivity and specificity provided by the manufacturer were >99% and 99.3% respectively[69].

Serum samples were also tested using the Rose-Bengal test (RBPT) for the detection of antibodies against *Brucella* antigen supplied by Jovac, Amman, Jordan. Briefly, a sample of serum (0.05 ml) was mixed with 0.05 ml of antigen on a microscope slide to produce a zone approximately 2 cm in diameter. The mixture was agitated gently for four minutes at room temperature and observed for agglutination, the sample being considered positive where a visible reaction was observed. Sensitivity and specificity were assumed to be 87.5% and 100% respectively[70]. (Combined performance values, positive on either RBPT or ELISA being calculated as 90% and 99.3% respectively).

**Outcomes.** The primary outcome was *Brucella* seropositive status on either RBPT or ELISA. Secondary outcomes were potential associations between possible risk factors and seropositive status.

**Statistical analysis.** Sample size calculations were based on an expected seroprevalence of 4% among livestock-owning household members, a 4:1 ratio of COH members to NCOH, 80% power, with 90% confidence level, and a design effect of 1·25. The target sample size was calculated as 946 individuals: 757 COH members and 189 NCOH members. 160 COH (40 per region) and 40 NCOH (10 per region) were randomly selected for potential inclusion, based on an estimated mean household sample size of 4·7, with additional households randomly selected (80 COH, 20 per region; and 20 NCOH, five per region) as required.

Seroprevalence estimates and their 95% CIs were calculated at the individual level for COHs and NCOHs, weighted by the number of COHs and NCOHs in each administrative region. The intraclass correlation coefficient was calculated to assess the extent to which seropositive individuals clustered within households. Cohen's $\kappa$ coefficient was calculated as a measure of agreement between ELISA and RBPT results.

Univariable associations between hypothetical risk factors and serological status were assessed by means of mixed-effects logistic regression, with household as a random effect and serological status as a binary outcome, without weighting for the region. Potential risk factors including temporospatial data, personal information, health and hygiene data and history of livestock contact during the previous 6 months, were analysed as categorical variables at the individual level (including the household-level variables of household size, dwelling type, water source, any home air-cooling system, number of livestock herds/flocks nearby, and all herd/flock-specific variables).

Variables with a p value of less than 0·20 in univariable analysis were considered for inclusion in the multivariable models, except for any variables missing more than 10% of values, which were examined separately[48]. Collinearities between variables were examined with use of the Pearson's coefficient, with a threshold of 0·4 or greater to define direct collinearity and −0·4 for inverse collinearity. Collinear variables were excluded from the same multivariable model and tested in separate models, except for a-priori variables (age and sex), which were included alongside any collinear variables in the same model.

Multivariable models were constructed using mixed-effects regression, with household as a random effect. Models were first constructed with a backwards stepwise method; the variable to be removed at each step was identified by comparing models with each of the non-a-priori variables removed and selecting the model with the lowest Akaike Information Criterion score, provided that removing the variable did not change the log odds ratio of any non-a-priori variables by more than 10%[48]. Models were then constructed with a forward stepwise method to ensure the same model was constructed by either method. Regression analysis confidence intervals were generated with the profile likelihood technique.

All statistical analyses were done in R (version 4.0.3) with mixed-effects models generated with use of the glmer function of the package lme4 (version 1.1–26).

### Reporting summary
Further information on research design is available in the Nature Portfolio Reporting Summary linked to this article.

## Data availability
All authors had full access to all data in the study. The data supporting the findings of this study is available upon reasonable request to the corresponding author, subject to approval by the relevant ethics committee, along with study questionnaires.

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

## Acknowledgements

We sincerely thank the Ministry of Health and Ministry of Agriculture in Jordan. Also Fares A. Altakhainah, Ghassab H. Hasanat, Hassan H. Alhusainat, Abdalmajeed M. Alajlouni, Suk Woo Lee, Yeong Kyoung Kang, and Hitesh Patel. This research was supported by a Medical Research Council Global Challenges Research Fund Foundation award (2017–2018, URN 2017 1735-3, JG), a Foreign and Commonwealth Office Bilateral Programme Budget Fund award (2013–2014, JG) from the British Embassy in Amman and the UK International Biosecurity Programme (within the context of a WOAH twinning initiative between the RVC and JUST).

## Author contributions

P.H., J.G., P.M., and J.C. designed the study, A.A., E.A., and W.H. facilitated fieldwork in Jordan, and T.H., P.H., and M.G. coordinated and carried out the field work. B.A. performed laboratory testing. M.G. managed the data, performed the statistical analysis, and generated figures, P.H., J.G., and S.N. participated in the statistical analysis. P.H. wrote the manuscript, M.G., J.G., P.M., J.C., I.M., I.P., T.H., and B.C. participated in writing the manuscript. All authors critically reviewed and commented on the manuscript.

## Competing interests

E.A. was project coordinator for the project "Reducing the threat of MERS-CoV and avian influenza in Jordan and strengthening regional disease surveillance capacity", supported by the US Department of Defence Threat Reduction Agency, Biological Threat Reduction Program. All other authors declare no competing interests.
