## [Peer Review file · Nature Communications]

Camel milk is a neglected source of brucellosis among rural Arab communities

Corresponding Author: Dr Peter Holloway

Version 0:

Reviewer comments:

Reviewer #1

(Remarks to the Author)

MS # NCOMMS-23-55379

Title: Camels and brucellosis: a neglected source of a neglected disease among rural Arab communities

Worldwide, brucellosis remains the most prevalent bacterial zoonosis. Consequently, any study done on the topic is highly beneficial. This is a really nice study. The author's conclusions are well-supported by the data in this well-written study. The manuscript's presentation is clear, its language level is appropriate, and its conclusions are backed up by the findings.

I consider that the presented work requires some modifications before it is suitable for publication.

- 1) To what degree do the authors conclude that the seropositivity in camels revealed in this study is not the result of vaccination? keeping in mind that vaccination is a common practice in certain countries nearby. It would be helpful if the authors could make it clear that the sample animals did not receive a brucellosis vaccination.
- 2) The introduction does not provide sufficient background and include all relevant references and some important publications are missing including the following:
 - 1- Al-Amr M, Abasi L, Khasawneh R, Almharat S, Al-Smadi R, Abbasi N, Rabadi O, Oudat R. Epidemiology of human brucellosis in military hospitals in Jordan: A five-year study. *J Infect Dev Ctries*. 2022 Dec 21;16(12):1870-1876. doi: 10.3855/jidc.16861. PMID: 36753655.
 - 2- Al-Ani F, Al Ghenaimi S, Hussein E, Al Mawly J, Al MushaiKi K, Al Kathery S. Human and animal brucellosis in the Sultanate of Oman: an epidemiological study. *J Infect Dev Ctries*. 2023 Jan 31;17(1):52-58. doi: 10.3855/jidc.17286. PMID: 36795918.
 - 3- Al-Marzooqi W, Elshafie EI, Al-Toobi A, Al-Hamrashdi A, Al-Kharousi K, El-Tahir H, Jay M, Corde Y, ElTahir Y. Seroprevalence and Risk Factors of Brucellosis in Ruminants in Dhofar Province in Southern Oman. *Vet Med Int*. 2022 Nov 4;2022:3176147. doi: 10.1155/2022/3176147. PMID: 36386268; PMCID: PMC9652075.
 - 4- Al-Majali AM, Al-Qudah KM, Al-Tarazi YH, Al-Rawashdeh OF. Risk factors associated with camel brucellosis in Jordan. *Trop Anim Health Prod*. 2008 Apr;40(3):193-200. doi: 10.1007/s11250-007-9080-7. PMID: 18484121.
 - 5- Abutarbush SM, Hamdallah A, Hawawshah M, Alsawalha L, Elizz NA, Dodeen R. Implementation of One Health approach in Jordan: Review and mapping of ministerial mechanisms of zoonotic disease reporting and control, and inter-sectoral collaboration. *One Health*. 2022 Jun 8;15:100406. doi: 10.1016/j.onehlt.2022.100406. PMID: 36277088; PMCID: PMC9582409.
 - 6- McAlester J, Kanazawa Y. Situating zoonotic diseases in peacebuilding and development theories: Prioritizing zoonoses in Jordan. *PLoS One*. 2022 Mar 17;17(3):e0265508. doi: 10.1371/journal.pone.0265508. PMID: 35298543; PMCID: PMC8929606.
 - 7- Abo-Shehada MN, Abu-Halaweh M. Risk factors for human brucellosis in northern Jordan. *East Mediterr Health J*. 2013 Feb;19(2):135-40. PMID: 23516823.
 - 8- Abo-Shehada MN, Odeh JS, Abu-Essud M, Abuharfeil N. Seroprevalence of brucellosis among high risk people in northern Jordan. *Int J Epidemiol*. 1996 Apr;25(2):450-4. doi: 10.1093/ije/25.2.450. PMID: 9119573.
 - 9- Abu Shaqra QM. Epidemiological aspects of brucellosis in Jordan. *Eur J Epidemiol*. 2000 Jun;16(6):581-4. doi: 10.1023/a:1007688925027. PMID: 11049102.
 - 10- Dajani YF, Masoud AA, Barakat HF. Epidemiology and diagnosis of human brucellosis in Jordan. *J Trop Med Hyg*. 1989 Jun;92(3):209-14. PMID: 2738993.
 - 11- Samadi A, Ababneh MM, Giadinis ND, Lafi SQ. Ovine and Caprine Brucellosis (*Brucella melitensis*) in Aborted Animals

in Jordanian Sheep and Goat Flocks. *Vet Med Int.* 2010 Oct 28;2010:458695. doi: 10.4061/2010/458695. PMID: 21052561; PMCID: PMC2971571.

12- Abo-Shehada MN, Rabi AZ, Abuharfeil N. The prevalence of brucellosis among veterinarians in Jordan. *Ann Saudi Med.* 1991 May;11(3):356-7. doi: 10.5144/0256-4947.1991.356. PMID: 17590745.

13- Issa H, Jamal M. Brucellosis in children in south Jordan. *East Mediterr Health J.* 1999 Sep;5(5):895-902. PMID: 10983528.

14- Al-Talafhah AH, Lafi SQ, Al-Tarazi Y. Epidemiology of ovine brucellosis in Awassi sheep in Northern Jordan. *Prev Vet Med.* 2003 Sep 12;60(4):297-306. doi: 10.1016/s0167-5877(03)00127-2. PMID: 12941554.

15- Aldomy FM, Jahans KL, Altarazi YH. Isolation of *Brucella melitensis* from aborting ruminants in Jordan. *J Comp Pathol.* 1992 Aug;107(2):239-42. doi: 10.1016/0021-9975(92)90040-2. PMID: 1452817.

3) Brucellosis is caused by brucella spp. Use the phrase "brucella" rather than "brucella spp." throughout the manuscript. For instance, seroprevalence of brucella infection rather than seroprevalence of brucella spp.

4) Was there a method or assumption utilized to calculate the total number of animals sampled?

5) Include the scientific name "camelus dromedarius" following the first mention of "dromedary camels."

6) In results, and under `Humans` for non-camel owning households add the acronym (NCOH)

7) There is confusion when using the terms Middle East and Eastern Mediterranean.

8) You might have to explain what is meant by "Islamic Hadith."

9) Blood samples were collected in `plain` serum vacutainer.

10) Spell out JUST, and MoH.

Reviewer #2

(Remarks to the Author)

The study reports results from a cross sectional study that estimated the prevalence of *Brucella* spp in camels and humans. The results obtained mirror those that have been published where the prevalence levels in livestock are always low while those in humans are quite high. The noteworthy findings given here is the high *Brucella* spp seroprevalence in people who come from camel-owning households compared to those that don't own camels.

I have a few concerns on the methodology and results provided.

1. It appears that the camels and human sampling activities were implemented in different households. If yes, why wasn't it possible to conduct a linked study where both human and livestock are recruited from the same household. That could have strengthened the results that compare COW and NCOH.

2. Camels sampled in the earlier stages of the study, i.e., February 2014 to December 2015 were sampled using non-probabilistic sampling methods while the rest of the animals sampled in September 2017 – October 2018 used probabilistic sampling techniques. Wouldn't combining these approaches introduce bias? The potential impact of this approach should be captured in the discussion

3. I think the discussion uses a few paragraphs to discuss the results. A large section of the discussion veers off to other discussion points that were not included in the study. I would have expected more discussion on the implications of the low prevalences observed in camels, the importance of some of the parameters generated in the results such as the ICC.

4. The methodology does not give a good description of all the types of data collected. One stumbles on variables used for risk factor analysis in the analysis section yet these are not introduced earlier. Variables collected like age, sex etc. should be highlighted while a description on methods used to collect data is introduced.

5. In the methodology section, it is stated that models were built using age as a continuous variable. An attempt should be made to discuss whether this variable met the linearity assumption. However in the results section, the tables given has age categorised as a factor variable. In the human models, the criterion used to include variables from univariable analysis into multivariable analysis of $p < 0.05$ is also very stringent. In many studies, $p < 0.20$ is often used

6. What is the justification for using 4:1 ratio for the selection of subjects for COH:NCOH comparison?

The conclusion for the difference in COH:NCOH is based largely on data collected from questionnaires. A linked study would have given stronger findings

Reviewer #3

(Remarks to the Author)

Reported key results:

Camels occupy a unique status in Jordan and the milk may be a transmitter of disease to humans if consumed raw. The authors attempt to estimate seroprevalence and identify potential risk factors by conducting a cross-sectional of humans and camels in two regions of the country. The authors claim seroprevalence among camels is low, but high seroprevalences were found in human populations, and that the results provided evidence of association between consumption of camel milk and human infection.

Validity:

Data interpretation:

- It seems like one of the co-authors published an article investigating cattle seroprevalence in Jordan (A.M. Al-Majali, et. al., 2009. Seroprevalence and risk factors for bovine brucellosis in Jordan), also sampling within the same two regions as this study finding cattle seroprevalence in Jordan = 6.5%; Ma'an = 30.7%; Aqaba = 0.5%).
 - o This mirrors the camel results from this study.
 - o If this is correct, camels could be infected with *B. abortus*, and or humans infected from cattle handling and/or milk consumption.
- The entire analysis should be conducted separately on Ma'an due to the low prevalence of camel brucellosis in Aqaba. Incorporating a location with low to no prevalence in livestock dilutes the human risk estimate.
- IDVet ID Screen® Brucellosis Serum Indirect Multi-species ELISA has been recommended for cattle, sheep, goats, and pigs by the manufacturer, but for the use in camels.
 - o Since the manufacturer does not overtly support its use in camels, the authors should have optimized, standardized, and set cut-off values per the World Organization of Animal Health's Terrestrial Manual. Chapter 3.1.4 § 2.5.1.1.3. Infection with *Brucella* in pigs and camelids: "In the absence of appropriate international standard sera the test should be duly validated and the cut-off established in the test population with appropriate validation techniques (see chapter 1.1.6)."
 - o Without validation of the test used, the reviewer cannot have confidence in the results or conclusions.

Conclusions:

- The conclusions are not supported by the results
- o "camels present an important source of zoonotic transmission due to widespread consumption of raw milk"
 - Infection could originate from cattle, and
 - There is no supporting evidence within the surveys that camel milk is consumed raw, or sheep/goat milk is boiled
- o "In the meantime, control measures among camels must focus on improved biosecurity, including closed herd management practices and regular testing of breeding males, together with improved control among the small ruminant populations – and with camels managed away from small ruminants where possible, particularly around parturition."
 - This conclusion is not supported by results, and
 - This statement is more of a discussion point than a conclusion

Significance:

- If adequately substantiated, the results 1) consumption of camel milk is associated with human brucellosis and 2) consumption of sheep and goat milk, along with handling sheep and goats is not associated with the disease, would be significant.
- Unfortunately, due to deficiencies in the methodology, these cannot be supported by the results

Data and methodology:

- Methods are not clear enough to reproduce results
- The authors need to indicate how:
 - o Sample sizes were calculated for humans, animals, herds, and animals per herd,
 - o Sensitivity and specificity of tests were estimated,
 - o Adjustment of seroprevalence was conducted especially due to the extremely low series sensitivity of the RBT/iELISA in camels (approximately 56.9%)
- A more informative description of the multivariate analysis is required,
- It appears that the surveys assume that sheep milk is being boiled and camel milk is being consumed raw. There needs to be data derived from the surveys to support these assertions.
- The study should have assessed consumption of raw milk, not just milk in general,
- The reviewer was not able to ascertain whether the authors assessed for a difference in camel ownership by region

Suggested improvements:

- the authors need to have a professional editor review the paper for mistakes,
- Conduct data analysis solely within Ma'an,
- Draft the methods in a format that the reader may reproduce the results,
- Multivariate analysis needs to be explained with more clarity and detail,
- Check that cited information is accurate, up to date, put into the correct context, and the information is cited correctly

Clarity and context:

- The methods are not clear enough to reproduce results, and
- Results have not been provided with sufficient context and consideration of previous work, as a previous study in cattle changes the interpretation of this investigation

References:

- There are many statements throughout the entire manuscript that are either not substantiated by the research or require citation.
- There are a few statements that cite outdated information. For example: "With almost a million human cases estimated to occur each year worldwide, and with over half of these occurring within the Eastern Mediterranean region, the World Health Organization (WHO) has described brucellosis as being one of the world's leading zoonotic diseases, and a neglected disease"
 - o New models estimate millions of new cases per year (C.G. Laine, et. al., 2023, Global Estimate of Human Brucellosis Incidence)
 - o The Eastern Mediterranean reports about 15% of the total human cases (Information available from WOA-H-WAHIS)
- Authors need to check that information is accurate, up to date, put into the correct context, and cited correctly throughout the

manuscript.

Version 1:

Reviewer comments:

Reviewer #1

(Remarks to the Author)

Even though you have addressed most of my concerns in a satisfactory manner, further comments and clarifications are still required.

1) To what degree do the authors conclude that the seropositivity in camels revealed in this study is not the result of vaccination? keeping in mind that vaccination is a common practice in certain countries nearby. It would be helpful if the authors could make it clear that the sample animals did not receive a brucellosis vaccination.

Author reply: We thank the reviewer for their comment and have addressed this point in line 96 of the manuscript. None of the camels included in the study have been vaccinated against brucellosis. Information regarding vaccination of camels was included in the camel questionnaire data, and these results have now been added to the manuscript, to clarify this point.

Reviewer's comment: I'm not happy with this response. Because you believe there are no approved brucella vaccinations for camels, you presumed that the sampled camels had not had a vaccination (lines 28 and 282-283). This unsubstantiated assertion should be removed from the text since it causes confusion. Brucella vaccinations, both inactivated and attenuated, have been effectively administered to camels (see Wernery U. Camelid brucellosis: a review. Rev Sci Tech. 2014 Dec;33(3):839-57). Camel vaccination against brucellosis is a widespread practice in Saudi Arabia, a neighboring country. Consequently, it's possible that some of the sampled camels had already received vaccinations. As a result, the inclusion criteria ought to make it very apparent that only camels that had not had a vaccination were sampled. It is insufficient to state in paragraph three, line 95, that "No camels had prior history of vaccination against Brucella." In addition, am interested in learning if the discussion paragraph (lines 292-295) on the reported intake of the Rev 1 vaccine suggests that the vaccine is widely used to prevent brucellosis in camels in the study area.

2) The introduction does not provide sufficient background and include all relevant references and some important publications are missing including the following:

Author reply: The introduction has been rewritten to address this point. We thank the reviewer for the published literature highlighted. We have now included reference to these studies within the introduction and discussion sections of the manuscript.

Reviewer's comment: Yes, all suggested articles have been added to the manuscript except Abu Shaqra QM, 2000.

3) Throughout the text, the full stop (.) is not inserted correctly after a sentence or incorrectly preceding the reference. Most pages have this major grammatical error, which is particularly noticeable in the introduction and discussion (examples: lines 47-48, 50, 54, 56-67, 59-60, 62, 63-64, 69, 73, 76, 108, 240, 243, 246, 262, 266, 275, 277, 280, 282, 287, 291, 305-306, 322, 331, 466, 478, 575).

4) References 1 (spell out WHO), 6 (small letters), 37 (WOAH now), 51 (pages, spell out APAHP), 63, 68 need to be revised according to the journal's rules.

Kindly address each comment point by point and include the exact line where you made the proposed changes.

Good luck,

Reviewer #2

(Remarks to the Author)

I have gone through the comments that were made by the reviewers and evaluated whether the authors have addressed them. I am happy to note that good responses have been given. There are a few minor corrections that need to be made:

1. The description of the questionnaires that were used to collect the data (lines 406 – 407) need to also specify that data on vaccination status of the animals sampled were collected. The authors mention that they asked for this information but this is not stated in text

2. The authors had been asked to delete "app" after Brucella but this is still found in Lines 450, 523

Otherwise I don't have more comments.

REVIEWER COMMENTS

Reviewer #1 (Remarks to the Author):

MS # NCOMMS-23-55379

Title: Camels and brucellosis: a neglected source of a neglected disease among rural Arab communities

Worldwide, brucellosis remains the most prevalent bacterial zoonosis. Consequently, any study done on the topic is highly beneficial. This is a really nice study. The author's conclusions are well-supported by the data in this well-written study. The manuscript's presentation is clear, its language level is appropriate, and its conclusions are backed up by the findings.

I consider that the presented work requires some modifications before it is suitable for publication.

- 1) To what degree do the authors conclude that the seropositivity in camels revealed in this study is not the result of vaccination? keeping in mind that vaccination is a common practice in certain countries nearby. It would be helpful if the authors could make it clear that the sample animals did not receive a brucellosis vaccination.

We thank the reviewer for their comment and have addressed this point in line 96 of the manuscript. None of the camels included in the study have been vaccinated against brucellosis. Information regarding vaccination of camels was included in the camel questionnaire data, and these results have now been added to the manuscript, to clarify this point.

- 2) The introduction does not provide sufficient background and include all relevant references and some important publications are missing including the following:

The introduction has been rewritten to address this point. We thank the reviewer for the published literature highlighted. We have now included reference to these studies within the introduction and discussion sections of the manuscript.

1-Al-Amr M, Abasi L, Khasawneh R, Almharat S, Al-Smadi R, Abbasi N, Rabadi O, Oudat R. Epidemiology of human brucellosis in military hospitals in Jordan: A five-year study. *J Infect Dev Ctries*. 2022 Dec 21;16(12):1870-1876. doi: 10.3855/jidc.16861. PMID: 36753655.

2- Al-Ani F, Al Ghenaimi S, Hussein E, Al Mawly J, Al MushaiKi K, Al Kather, Human and animal brucellosis in the Sultanate of Oman: an epidemiological study. *J Infect Dev Ctries*. 2023 Jan 31;17(1):52-58. doi: 10.3855/jidc.17286. PMID: 36795918.

3- Al-Marzooqi W, Elshafie EI, Al-Toobi A, Al-Hamrashdi A, Al-Kharousi K, El-Tahir H, Jay M, Corde Y, EITahir Y. Seroprevalence and Risk Factors of Brucellosis in Ruminants in Dhofar Province in Southern Oman. *Vet Med Int*. 2022 Nov 4;2022:3176147. doi: 10.1155/2022/3176147. PMID: 36386268; PMCID: PMC9652075.

- 4- Al-Majali AM, Al-Qudah KM, Al-Tarazi YH, Al-Rawashdeh OF. Risk factors associated with camel brucellosis in Jordan. *Trop Anim Health Prod.* 2008 Apr;40(3):193-200. doi: 10.1007/s11250-007-9080-7. PMID: 18484121.
- 5- Abutarbush SM, Hamdallah A, Hawawsheh M, Alsawalha L, Elizz NA, Dodeen R. Implementation of One Health approach in Jordan: Review and mapping of ministerial mechanisms of zoonotic disease reporting and control, and inter-sectoral collaboration. *One Health.* 2022 Jun 8;15:100406. doi: 10.1016/j.onehlt.2022.100406. PMID: 36277088; PMCID: PMC9582409.
- 6- McAlester J, Kanazawa Y. Situating zoonotic diseases in peacebuilding and development theories: Prioritizing zoonoses in Jordan. *PLoS One.* 2022 Mar 17;17(3):e0265508. doi: 10.1371/journal.pone.0265508. PMID: 35298543; PMCID: PMC8929606.
- 7- Abo-Shehada MN, Abu-Halaweh M. Risk factors for human brucellosis in northern Jordan. *East Mediterr Health J.* 2013 Feb;19(2):135-40. PMID: 23516823.
- 8- Abo-Shehada MN, Odeh JS, Abu-Essud M, Abuharfeil N. Seroprevalence of brucellosis among high risk people in northern Jordan. *Int J Epidemiol.* 1996 Apr;25(2):450-4. doi: 10.1093/ije/25.2.450. PMID: 9119573.
- 9- Abu Shaqra QM. Epidemiological aspects of brucellosis in Jordan. *Eur J Epidemiol.* 2000 Jun;16(6):581-4. doi: 10.1023/a:1007688925027. PMID: 11049102.
- 10- Dajani YF, Masoud AA, Barakat HF. Epidemiology and diagnosis of human brucellosis in Jordan. *J Trop Med Hyg.* 1989 Jun;92(3):209-14. PMID: 2738993.
- 11- Samadi A, Ababneh MM, Giadinis ND, Lafi SQ. Ovine and Caprine Brucellosis (*Brucella melitensis*) in Aborted Animals in Jordanian Sheep and Goat Flocks. *Vet Med Int.* 2010 Oct 28;2010:458695. doi: 10.4061/2010/458695. PMID: 21052561; PMCID: PMC2971571.
- 12- Abo-Shehada MN, Rabi AZ, Abuharfeil N. The prevalence of brucellosis among veterinarians in Jordan. *Ann Saudi Med.* 1991 May;11(3):356-7. doi: 10.5144/0256-4947.1991.356. PMID: 17590745.
- 13- Issa H, Jamal M. Brucellosis in children in south Jordan. *East Mediterr Health J.* 1999 Sep;5(5):895-902. PMID: 10983528.
- 14- Al-Talafhah AH, Lafi SQ, Al-Tarazi Y. Epidemiology of ovine brucellosis in Awassi sheep in Northern Jordan. *Prev Vet Med.* 2003 Sep 12;60(4):297-306. doi: 10.1016/s0167-5877(03)00127-2. PMID: 12941554.
- 15- Aldomy FM, Jahans KL, Altarazi YH. Isolation of *Brucella melitensis* from aborting ruminants in Jordan. *J Comp Pathol.* 1992 Aug;107(2):239-42. doi: 10.1016/0021-9975(92)90040-2. PMID: 1452817.

- 3) Brucellosis is caused by brucella spp. Use the phrase "brucella" rather than "brucella spp." throughout the manuscript. For instance, seroprevalence of brucella infection rather than seroprevalence of brucella spp.

The manuscript has now been edited to address this point.

- 4) Was there a method or assumption utilized to calculate the total number of animals sampled?

We have revised the manuscript to address this point by including a description of camel sample size calculation in the methods (lines 444–449 and lines 544–550).

- 5) Include the scientific name "camelus dromedarius" following the first mention of "dromedary camels."

We have revised the manuscript to address this point (Line 28 & 71)

- 6) In results, and under `Humans` for non-camel owning households add the acronym (NCOH)

We have revised the manuscript to address this point (Line 130)

- 7) There is confusion when using the terms Middle East and Eastern Mediterranean.

We have revised the manuscript to address this point, by referring to the Middle East only, and using the term.

- 8) You might have to explain what is meant by "Islamic Hadith."

We have revised the manuscript to address this point (Line 236)

- 9) Blood samples were collected in `plain` serum vacutainer.

We have revised the manuscript to address this point (Line 409 and 519)

- 10) Spell out JUST, and MoH.

We have revised the manuscript to address this point (Line 380)

Reviewer #2 (Remarks to the Author):

The study reports results from a cross sectional study that estimated the prevalence of Brucella spp in camels and humans. The results obtained mirror those that have been published where the prevalence levels in livestock are always low while those in humans are quite high. The noteworthy findings given here is the high Brucella

app seroprevalence in people who come from camel-owning households compared to those that don't own camels.

I have a few concerns on the methodology and results provided.

1. It appears that the camels and human sampling activities were implemented in different households. If yes, why wasn't it possible to conduct a linked study where both human and livestock are recruited from the same household. That could have strengthened the results that compare COW and NCOH.

This was a linked study with 131/172 COH (76.2%) of included COH households also having camels from their herd sampled. The manuscript has been clarified to reflect this point (Line 91&92 and Line 132&133) and included in the study profile (Figure 1). Ownership of a herd with at least one positive sampled camel is now presented as a variable in the univariable table (Table 6).

2. Camels sampled in the earlier stages of the study, i.e., February 2014 to December 2015 were sampled using non-probabilistic sampling methods while the rest of the animals sampled in September 2017 – October 2018 used probabilistic sampling techniques. Wouldn't combining these approaches introduce bias? The potential impact of this approach should be captured in the discussion.

There is indeed potential for the introduction of bias in camel seroprevalence estimates, due to the non-probabilistic sampling used in the period 2014-15. As a result of sampling the clientele of a local veterinary practice, this is likely to have led to an increased inclusion of herds belonging to owners seeking veterinary interventions. This has the potential for both upward and downward bias, due to owners of brucella infected herds potentially being more likely to seek veterinary interventions (due to reproductive losses), while disease-free herd owners are also potentially more likely to seek veterinary interventions, as a part of improved management practices. We have acknowledged this as a limitation in the discussion section of the manuscript (Line 343-347) and we are presenting seroprevalence values for each period individually. However, we do not believe this has had an important impact on seroprevalence estimates in camels, with seroprevalence among randomly selected camel herds broadly similar to those among non-probabilistically sampled herds, across the study period. With regard to association between seropositive status and potential risk factors, while herds from the 2014-15 period were not sampled probabilistically, we believe there are no obvious confounders that could have been distributed systematically different between study herds and the reference population and were not considered in our analysis.

3. I think the discussion uses a few paragraphs to discuss the results. A large section of the discussion veers off to other discussion points that were not included in the study. I would have expected more discussion on the implications of the low prevalences observed in camels, the importance of some of the parameters generated in the results such as the ICC.

We have also revised the manuscript to include more discussion of the implications of the low prevalence observed in camels (Lines 261-263) and other results, including ICC (Lines 340-342) that had previously been underrepresented in the discussion.

4. The methodology does not give a good description of all the types of data collected. One stumbles on variables used for risk factor analysis in the analysis section yet these are not introduced earlier. Variables collected like age, sex etc. should be highlighted while a described on methods used to collect data is introduced.

We have revised the methods section of the manuscript to include description of types of data collected, including questions regarding personal information, livestock engagement, consumption of livestock products and hygiene practices (Lines 509-512).

5. In the methodology section, it is stated that models were built using age as a continuous variable. An attempt should be made to discuss whether this variable met the linearity assumption. However in the results section, the tables given has age categorised as a factor variable.

In the human models, the criterion used to include variables from univariable analysis into multivariable analysis of $p < 0.05$ is also very stringent. In many studies, $p < 0.20$ is often used

Thank you for the suggestions.

The methods section has been revised to address this point, with age presented as categorical variable in human descriptive and univariate statistics, and as a binary variable ($>$ median) in camel descriptive and univariate statistics.

Regarding inclusion of variables $p < 0.2$ cut-off, we have now revised the methods accordingly.

6. What is the justification for using 4:1 ration for the selection of subjects for COH:NCOH comparison?

The selection of subjects using a 4:1 ratio COH : NCOH is based upon study design aimed at capturing potential risk factors for brucellosis associated with camel ownership. NCOHs were included as a reference group to assess potential differences in brucella seroprevalence between household types, but in the interest of the study objectives concerning estimation of prevalence in camels and ascertainment of risk factors associated with camel presence in the household the ratio of COH to NCOH was increased.

The conclusion for the difference in COH:NCOH is based largely on data collected from questionnaires. A linked study would have given stronger findings.

As mentioned above, the study is in fact linked, apologies for not having made this clear. The manuscript has been clarified to reflect this point (Line 91&92 and Line 132&133), with the following data presented within the descriptive and univariate statistics; among households members owning camels tested for *Brucella* serological status, seroprevalence among individuals drinking raw camel milk from herds where a camel had tested positive was 50.0% (2/4), compared to 15.7% (44/281) among

members drinking raw milk from herds where all camels tested had been seronegative (OR_{min.adj} 9.44 [0.36–244.96], p=0.18).

Reviewer #3 (Remarks to the Author):

Reported key results:

Camels occupy a unique status in Jordan and the milk may be a transmitter of disease to humans if consumed raw. The authors attempt to estimate seroprevalence and identify potential risk factors by conducting a cross-sectional of humans and camels in two regions of the country. The authors claim seroprevalence among camels is low, but high seroprevalences were found in human populations, and that the results provided evidence of association between consumption of camel milk and human infection.

Validity:

Data interpretation:

- It seems like one of the co-authors published an article investigating cattle seroprevalence in Jordan (A.M. Al-Majali, et. al., 2009. Seroprevalence and risk factors for bovine brucellosis in Jordan), also sampling within the same two regions as this study finding cattle seroprevalence in Jordan = 6.5%; Ma'an = 30.7%; Aqaba = 0.5%).
 - o This mirrors the camel results from this study.
 - o If this is correct, camels could be infected with *B. abortus*, and or humans infected from cattle handling and/or milk consumption.

The study by A.M. Al-Majali, et. al., 2009, described by the reviewer, omits information regarding the number of cattle present in Ma'an and Aqaba regions, when presenting sample prevalences, with OIE data for Jordan listing 10 cattle in Aqaba and 116 in Ma'an, out a total population of 42,990 cattle nationwide). In summary, the livestock populations in southern Jordan are entirely dominated by camels, sheep and goats (due to the arid environment) and out of the 203 households included in the study, none owned cattle. The manuscript has been revised to clarify this point (Line 137-139) with the absence of cattle also described in the descriptive table (Table 4).

<https://rr-middleeast.woah.org/en/about-us/regional-members-of-woah/jordan/#:~:text=Jordan%20had%20about%2035%2C000%20head,planned%20to%20increase%20their%20numbers>

- The entire analysis should be conducted separately on Ma'an due to the low prevalence of camel brucellosis in Aqaba. Incorporating a location with low to no prevalence in livestock dilutes the human risk estimate.

We thank the reviewer for their comment and in order to make clear to readers the differences between the two areas we are presenting separately the results form both areas. In order to assess the hypothesis that camels represent a risk factor for infection both areas are combined adjusting for the potential confounding effect of the area.

Apparent camel seroprevalences in Ma'an was higher than that in Aqaba, being 2.4% (8/340) and 0.4% (2/544) respectively OR 6.53 [1.62–43.44], $p=0.0069$). Adjusted herd-level seroprevalences (weighted for region and adjusted for test performance values and herd sampling proportion) being 30.6% (95% CI 23.5–38.3) in Ma'an and 4.7% (95% CI 2.1–8.2) in Aqaba.

Likewise, apparent seroprevalence among members of camel-owning households (COH) was higher in Ma'an than in Aqaba; being 17.9% (68/379) in Ma'an and 3.8% (11/288) in Aqaba, (OR 5.51 [2.85–10.62], $p<0.0001$).

Among non-camel owning households (NCOH) however, apparent seroprevalence was relatively similar in both regions; 7.1% (7/99) in Ma'an and 5.0% (5/101) in Aqaba (OR 1.46 [0.45–4.77], $p=0.55$), suggesting the difference in seroprevalences among COH members in Ma'an and Aqaba is likely attributable to the difference in camel seroprevalence observed between the two regions, with zoonotic transmission occurring via consumption of raw milk. This is supported by the evidence that history of drinking camel milk (in the last 6 months) in Ma'an is associated with risk of seropositive status (OR 3.87 [1.94–7.72], $p=0.0001$), while a history of drinking camel milk in Aqaba is not, (OR 1.53 [0.37–6.38], $p=0.56$).

In response to comments from the reviewer we re-examined the human data, stratifying for Ma'an only, and found results to be largely consistent with those observed for the two regions combined. Region is now included as a covariate in our multivariable analysis, to adjust for potential confounding.

- IDVet ID Screen® Brucellosis Serum Indirect Multi-species ELISA has been recommended for cattle, sheep, goats, and pigs by the manufacturer, but for the use in camels.

o Since the manufacturer does not overtly support its use in camels, the authors should have optimized, standardized, and set cut-off values per the World Organization of Animal Health's Terrestrial Manual. Chapter 3.1.4 § 2.5.1.1.3. Infection with *Brucella* in pigs and camelids: "In the absence of appropriate international standard sera the test should be duly validated and the cut-off established in the test population with appropriate validation techniques (see chapter 1.1.6)."

o Without validation of the test used, the reviewer cannot have confidence in the results or conclusions.

We thank the reviewer for raising this issue. Due to lack of validation of the IDVet ID Screen® Brucellosis Serum Indirect Multi-species ELISA, we have now used the Complement Fixation Test (CFT) as a confirmatory test following RBPT screening, in compliance with OIE guidelines, when presenting individual and herd-level prevalences.

However, iELISA results do show substantial agreement CFT results, with a Kappa coefficient of 0.76 (95% CI 0.51–1.00), among 33 positive samples on RBPT (from 884 camel serum samples tested); seropositivity to either test being used for purposes of logistic regression analysis, due to the singularity of models when using CFT confirmed positives alone, due to low prevalence.

	CFT -ve	CFT +ve	Total
ELISA -ve	23	2	25
ELISA +ve	1	7	8
Total	24	9	33

Conclusions:

- The conclusions are not supported by the results
 - o “camels present an important source of zoonotic transmission due to widespread consumption of raw milk”
 - ♣ Infection could originate from cattle, and
 - ♣ There is no supporting evidence within the surveys that camel milk is consumed raw, or sheep/goat milk is boiled

As mentioned previously, no households in the sample population owned cattle, with cattle being almost entirely absent from the south of Jordan due to arid environmental conditions and a low resource setting (occasionally being present very briefly prior to slaughter at local abattoirs). We have revised the manuscript to clarify this point Line 137-139) and adding cattle ownership status (absence of) to the descriptive table (Table 4).

With regard to evidence to support camel milk being consumed raw, in a follow-up survey conducted among 69/172 (40.1%) of COH, including 503 members, 308 individuals reported history of consuming camel milk (more than never), of whom 92.5% (285/308) stated that they always consumed it raw, and 320 individuals reported history of consuming small ruminant milk (more than never), of whom 13.1% stated they occasionally (more than rarely) drank it raw).(Lines 200-204)

- o “In the meantime, control measures among camels must focus on improved biosecurity, including closed herd management practices and regular testing of breeding males, together with improved control among the small ruminant populations – and with camels managed away from small ruminants where possible, particularly around parturition.”
 - ♣ This conclusion is not supported by results, and
 - ♣ This statement is more of a discussion point than a conclusion

We thank the reviewer for their comments, Study conclusions regarding potential brucella control measures among camels have been revised to reflect this point (Lines 348–360).

Significance:

- If adequately substantiated, the results 1) consumption of camel milk is associated with human brucellosis and 2) consumption of sheep and goat milk, along with handling sheep and goats is not associated with the disease, would be significant.
- Unfortunately, due to deficiencies in the methodology, these cannot be supported by the results

We have now revised our methods to incorporate comments received from the reviewers, increasing the threshold for inclusion of in multivariable models from $p=0.05$ to $p=0.20$ in univariable analysis, and including region as a covariate within multivariable models. Revised findings show birthing small ruminants (\geq weekly, in season) as a significant risk factor for *Brucella* seropositive status (OR_{adj} 1.86 [1.00–3.38], $p=0.048$), as would be expected in endemic setting, with weak evidence of consumption of daily raw small ruminant milk being associated with seropositive status (OR_{adj} 1.96 [0.95–3.93]). We believe that the revised description of the study methods (included revised statistical methods) provides robust evidence of association between daily consumption of raw camel milk with seropositive status (OR_{adj} 2.19 [1.23–3.94]) in southern Jordan.

Data and methodology:

- Methods are not clear enough to reproduce results
- The authors need to indicate how:
 - o Sample sizes were calculated for humans, animals, herds, and animals per herd,
 - o Sensitivity and specificity of tests were estimated,
 - o Adjustment of seroprevalence was conducted especially due to the extremely low series sensitivity of the RBT/iELISA in camels (approximately 56.9%)

The methods section of the manuscript has been revised to improve clarity and potential reproducibility of results, particularly regarding, i) sample size calculations (lines 444–449 & 544–549), ii) estimation of serological test performance, and iii) adjustment of seroprevalence estimates for camels, accounting for test performance values, with positivity on RBPT confirmed by CFT now being used for seroprevalence calculations, in place of confirmation in iELISA. RBPT performance values in camels used for purpose of adjustment were $Se = 86.7\%$, $Sp = 99.1\%$, based on Elsohaby et al, ([10.1016/j.prevetmed.2022.105771](https://doi.org/10.1016/j.prevetmed.2022.105771)) (line 419 & 420). CFT performance values in camels were assumed to be $Se = 99.0\%$, $Sp=98.4\%$, based on Khan et al ([10.5958/2277-8934.2016.00036.9](https://doi.org/10.5958/2277-8934.2016.00036.9)) (line 425), with combined performance values thus being calculated as being 85.8% and $>99.9\%$ (line 426&427).

- A more informative description of the multivariate analysis is required,

We have revised the methods to include a more detailed description of the logistic regression analyses used in the camel (461–482) and human (556–581) studies.

- It appears that the surveys assume that sheep milk is being boiled and camel milk is being consumed raw. There needs to be data derived from the surveys to support these assertions.

As mentioned above, in a follow-up survey conducted among 69/172 (40.1%) of COH, including 503 members, 308 individuals reported history of consuming camel milk (more than never), of whom 92.5% (285/308) stated that they always consumed it raw, and 320 individuals reported history of consuming small ruminant milk (more than never), of whom 13.1% stated they occasionally (more than rarely) drank it raw) (Lines 200-204).

- The study should have assessed consumption of raw milk, not just milk in general,

The wording of the manuscript has been revised to clarify that it was indeed only consumption of raw milk which was assessed. The questions being asked, ‘How often do you consume raw camel milk?, ‘How often do you consume raw milk from sheep and goats?’, ‘How often do you consume dairy products made from raw from sheep and goat milk? (Tables 7 & 8)

- The reviewer was not able to ascertain whether the authors assessed for a difference in camel ownership by region.

MoA records show 3089 livestock owning households in Ma’an, of which 317/3089 (10.3%) own camels, and 1359 livestock owning households in Aqaba, of which 265/1389 (19.1%) own camels, with 582/4478 (13.0%) owning camels overall. The manuscript has been amended to address this point (Lines 134-137).

Suggested improvements:

- the authors need to have a professional editor review the paper for mistakes,
- Conduct data analysis solely within Ma’an,
- Draft the methods in a format that the reader may reproduce the results,
- Multivariate analysis needs to be explained with more clarity and detail,
- Check that cited information is accurate, up to date, put into the correct context, and the information is cited correctly

As suggested by the reviewer, our revised manuscript has been carefully reviewed and edited prior to resubmission to check for mistakes. In addition, as suggested the methods section has been drafted in a revised format to facilitate reproducibility of the results for the reader and description of multivariable analysis methods has been revised to add clarity. Cited information has been revised and corrected to ensure that it is current, accurate and in context.

Regarding conducting data analysis in Ma’an only, as mentioned above, we believe the contrast in seroprevalences (human and camel) between Ma’an and Aqaba is helpful in demonstrating evidence of the zoonotic threat posed by camels from *Brucella*. Significantly higher camel seroprevalences in Ma’an, compared to Aqaba, (OR 6.53 [1.62–43.44], $p=0.0069$), are also reflected in higher human seroprevalences in Ma’an (among COH members), compared to Aqaba (OR 5.51 [2.85–10.62], $p<0.0001$); drinking raw camel milk being associated with seropositive status in Ma’an (OR 3.87 [1.94–7.72], $p=0.0001$), though not in Aqaba, (OR 1.53 [0.37–6.38], $p=0.56$), suggesting raw camel milk as a route of zoonotic *Brucella* transmission in Ma’an. These findings highlight the potentially important protective effect of reducing *Brucella* seroprevalence in camels among camel owning communities, particularly in view of the deep sociocultural challenges that exist in encouraging behavioural change regarding boiling of camel milk.

Clarity and context:

- The methods are not clear enough to reproduce results, and
- Results have not been provided with sufficient context and consideration of previous work, as a previous study in cattle changes the interpretation of this investigation

As mentioned earlier, the methods section has been revised to improve clarity and reproducibility of results. In addition, presentation of results has been revised to include greater context (for example, describing the absence of cattle among the study population (Lines 137-139)).

References:

- There are many statements throughout the entire manuscript that are either not substantiated by the research or require citation.
- There are a few statements that cite outdated information. For example: "With almost a million human cases estimated to occur each year worldwide, and with over half of these occurring within the Eastern Mediterranean region, the World Health Organization (WHO) has described brucellosis as being one of the world's leading zoonotic diseases, and a neglected disease"
 - o New models estimate millions of new cases per year (C.G. Laine, et. al., 2023, Global Estimate of Human Brucellosis Incidence)
 - o The Eastern Mediterranean reports about 15% of the total human cases (Information available from WOA-H-WAHIS)
- Authors need to check that information is accurate, up to date, put into the correct context, and cited correctly throughout the manuscript.

The manuscript has been revised to ensure that all statements are clearly supported by the study findings themselves or substantiated by other correctly cited literature. We thank the reviewer for their comments regarding the citing of outdated or incorrect information and have revised all citations to ensure they are correct and up to date with current data and findings.

REVIEWER COMMENTS

Reviewer #1 (Remarks to the Author):

Even though you have addressed most of my concerns in a satisfactory manner, further comments and clarifications are still required.

1) To what degree do the authors conclude that the seropositivity in camels revealed in this study is not the result of vaccination? keeping in mind that vaccination is a common practice in certain countries nearby. It would be helpful if the authors could make it clear that the sample animals did not receive a brucellosis vaccination.

Author reply: We thank the reviewer for their comment and have addressed this point in line 96 of the manuscript. None of the camels included in the study have been vaccinated against brucellosis. Information regarding vaccination of camels was included in the camel questionnaire data, and these results have now been added to the manuscript, to clarify this point.

Reviewer's comment: I'm not happy with this response. Because you believe there are no approved brucella vaccinations for camels, you presumed that the sampled camels had not had a vaccination (lines 28 and 282-283). This unsubstantiated assertion should be removed from the text since it causes confusion. Brucella vaccinations, both inactivated and attenuated, have been effectively administered to camels (see Wernery U. Camelid brucellosis: a review. Rev Sci Tech. 2014 Dec;33(3):839-57). Camel vaccination against brucellosis is a widespread practice in Saudi Arabia, a neighboring country. Consequently, it's possible that some of the sampled camels had already received vaccinations. As a result, the inclusion criteria ought to make it very apparent that only camels that had not had a vaccination were sampled. It is insufficient to state in paragraph three, line 95, that "No camels had prior history of vaccination against Brucella."

In addition, am interested in learning if the discussion paragraph (lines 292-295) on the reported intake of the Rev 1 vaccine suggests that the vaccine is widely used to prevent brucellosis in camels in the study area.

We thank the reviewer for their comments and have addressed this point by removing the text referring to the absence of a licensed *Brucella* vaccine for use in camels (line 28 of the previous manuscript submission) and prior history of vaccination (lines 95 of the previous manuscript submission). Rather, we have clarified this point in the discussion (line 282–283 of the previous manuscript) by adding the following text, *'Camel herds in the study population had not been vaccinated against Brucella. Government-led vaccination programmes in Jordan do not include camels and private access to the Rev 1 vaccine is heavily restricted, with those herd owners surveyed reporting absence of vaccination in their herds. In addition, importing of camels from neighbouring countries, including Saudi Arabia (where Rev 1 vaccination may potentially have occurred) is prohibited unless for direct slaughter, suggesting the Brucella seroconversion identified in the study population is likely attributable to natural infection. (44, 50) (Line 614–634)* We have also added the following text to the introduction, *'In addition, the lack of a commercially available Brucella vaccine licensed for use in camels means that vaccination is limited to off-*

label use only and, while practiced in some countries, is not widespread across the region (including Jordan where government-led Brucella livestock vaccination programmes do not include camels, and access by the private sector is restricted). (37). (Line 207–211)

2) The introduction does not provide sufficient background and include all relevant references and some important publications are missing including the following:

Author reply: The introduction has been rewritten to address this point. We thank the reviewer for the published literature highlighted. We have now included reference to these studies within the introduction and discussion sections of the manuscript.

Reviewer's comment: Yes, all suggested articles have been added to the manuscript except Abu Shaqra QM, 2000.

The reference Abu Shaqra QM, 2000 has now also been included in the revised manuscript. (Line 99).

3) Throughout the text, the full stop (.) is not inserted correctly after a sentence or incorrectly preceding the reference. Most pages have this major grammatical error, which is particularly noticeable in the introduction and discussion (examples: lines 47-48, 50, 54, 56-67, 59-60, 62, 63-64, 69, 73, 76, 108, 240, 243, 246, 262, 266, 275, 277, 280, 282, 287, 291,305-306, 322, 331, 466, 478, 575).

The manuscript has now been ammended to address this point.

4) References 1 (spell out WHO), 6 (small letters), 37 (WOAH now), 51 (pages, spell out APAHP), 63, 68 need to be revised according to the journal's rules.

The manuscript has now been ammended to address this point.

Kindly address each comment point by point and include the exact line where you made the proposed changes.

Good luck,

Reviewer #2 (Remarks to the Author):

I have gone through the comments that were made by the reviewers and evaluated whether the authors have addressed them. I am happy to note that good responses have been given. There are a few minor corrections that need to be made:

1. The description of the questionnaires that were used to collect the data (lines 406 – 407) need to also specify that data on vaccination status of the animals sampled were collected. The authors mention that they asked for this information but this is not stated in text

We have ammended the manuscript to address this point, '*Blood samples were collected from selected camels alongside administration of a structured questionnaire regarding potential camel-level and herd-level risk factors for Brucella exposure during the previous year, including Brucella vaccination.*' (Line 848–850)

2. The authors had been asked to delete "app" after Brucella but this is still found in Lines 450, 523

The manuscript has now been ammended to address this point.

Otherwise I don't have more comments.